# Chromosome segregation synchrony in *S. pombe* is noise limited and arises without positive feedback

Wendi Williams[1], Kien Phan[1,2], Jing Chen[1,3,4], Stefan Legewie[5], Julia Kamenz[1,6], and Silke Hauf[1,3,6,7]

Anaphase is a key cell cycle transition that ensures faithful genome inheritance. At anaphase onset, sister chromatids separate abruptly and synchronously upon activation of the protease separase. Major cell cycle transitions often involve positive feedback, which contributes to their abruptness and irreversibility; however, whether such feedback is required for anaphase remains unclear. Here, we analyze sister chromatid separation dynamics in fission yeast using high-resolution live-cell imaging and computational modeling. We find that anaphase synchrony relies on fast degradation of the separase inhibitor securin but does not require separase-mediated positive feedback. Hence, sister chromatid separation, being inherently irreversible, may be one of the few major cell cycle transitions that can proceed without positive feedback. A stochastic model fitted to the data revealed that separation synchrony is limited by stochasticity resulting from small-number effects. Together, these results support a feedback-independent mechanism for anaphase onset and identify molecular noise as a fundamental constraint on its temporal precision.

## Introduction

The sudden, synchronous splitting of sister chromatids at anaphase is visually one of the most striking transitions in the cell cycle. Until anaphase, sister chromatids are held together by cohesin, a large protein complex that can topologically encircle two sister chromatids (i.e., create "cohesion") and also influences global chromosome architecture by creating DNA loops (Uhlmann, 2025; Ochs and Gerlich, 2026; Nasmyth and Haering, 2009; Makrantoni and Marston, 2018). Cohesin is enriched at centromeric regions but also found along chromosome arms (Tomonaga et al., 2000; Schmidt et al., 2009; Watanabe, 2005). The establishment of cohesion during DNA replication ensures the proper attachment of sister chromatids to opposite poles of the mitotic spindle, a prerequisite for the equal distribution of genetic material to the two daughter cells (Tanaka et al., 2000; Sonoda et al., 2001). Sister chromatid cohesion is irreversibly lost during anaphase due to the proteolytic cleavage of the cohesin subunit Scc1 (Rad21 in fission yeast) by the protease separase (Uhlmann et al., 1999; Uhlmann et al., 2000; Tomonaga et al., 2000; Hauf et al., 2001; Yu et al., 2025). Scc1/Rad21 cleavage releases the topological entrapment of sister chromatids, allowing them to move to opposite ends of the cell (Uhlmann et al., 2000; Pauli et al., 2008; Oliveira et al., 2010).

Separase activation depends on the ubiquitination and subsequent degradation of its inhibitor, securin, mediated by the anaphase-promoting complex (APC/C) and the proteasome (Funabiki et al., 1996; Ciosk et al., 1998; Peters, 2006).

Across various organisms, chromosome separation occurs within a narrow time window, even though securin degradation proceeds much more slowly. For example, in human cells, complete securin degradation takes about 20 min, whereas the segregation of >40 chromosomes is completed within just 1–2 min (Hagting et al., 2002; Armond et al., 2019, *Preprint*; Sen et al., 2021). Similar patterns are observed in mouse oocytes (McGuinness et al., 2009; Thomas et al., 2021) and budding yeast (Lyons and Morgan, 2011; Lu et al., 2014). To explain the abrupt onset of sister chromatid separation, it has been proposed that separase activity increases in a switch-like manner (Holt et al., 2008; Shindo et al., 2012; Yaakov et al., 2012; Hellmuth et al., 2014). Supporting this idea, cohesin cleavage has been observed to rise sharply just before sister chromatids separate (Shindo et al., 2012; Yaakov et al., 2012). Such a switch-like increase in separase activity could result from positive feedback regulation, a common feature of major cell cycle transitions (Kapuy et al., 2009; Ferrell, 2013). In budding yeast, a positive feedback loop

---

[1]Department of Biological Sciences, Virginia Tech, Blacksburg, VA, USA;   [2]Interdisciplinary PhD Program in Genetics, Bioinformatics, and Computational Biology, Virginia Tech, Blacksburg, VA, USA;   [3]Center for the Mathematics of Biosystems, Virginia Tech, Blacksburg, VA, USA;   [4]Center for Soft Matter and Biological Physics, Virginia Tech, Blacksburg, VA, USA;   [5]Institute of Molecular Biology (IMB), Mainz, Germany;   [6]Friedrich Miescher Laboratory of the Max Planck Society, Tübingen, Germany;   [7]Fralin Life Sciences Institute, Virginia Tech, Blacksburg, VA, USA.

Correspondence to Silke Hauf: silke@vt.edu;   Julia Kamenz: j.l.kamenz@rug.nl

S. Legewie current affiliation is Institute of Biomedical Genetics, University of Stuttgart, Germany.   J. Kamenz's current affiliation is Molecular Systems Biology, Groningen Biomolecular Sciences and Biotechnology Institute, University of Groningen, Netherlands.

was proposed in which separase enhances securin degradation by indirectly promoting dephosphorylation of securin at Cdk1-dependent sites and accelerating securin ubiquitination (Holt et al., 2008). However, subsequent studies found that non-phosphorylatable and wild-type (WT) securin are degraded at similar rates and that the rate of cohesin cleavage is not strongly affected by expression of non-phosphorylatable securin (Yaakov et al., 2012; Lu et al., 2014). In human cells, PP2A-mediated dephosphorylation of separase-bound securin appears to decelerate rather than accelerate its degradation (Hellmuth et al., 2014; McGuinness et al., 2009). Thus, it remains unclear whether separase-mediated feedback is physiologically important and functionally conserved across eukaryotes.

To investigate the mechanisms behind sister chromatid separation synchrony, we implemented live-cell imaging with high temporal resolution in fission yeast, *Schizosaccharomyces pombe*. Fission yeast makes for a good model, as it has regional centromeres enriched in cohesin, similar to metazoan cells; only has three chromosomes that differ in size; and anaphase can be perturbed genetically or pharmacologically (Tong et al., 2019; Schmidt et al., 2009; Hayles and Nurse, 2018). Using live-cell microscopy, we quantified the timing and kinetics of sister chromatid separation and securin degradation in WT cells and analyzed how different perturbations alter these dynamics. We find that the synchrony of sister chromatid separation correlates with the speed of securin degradation. Combining these quantitative results with computational models suggests that separase-mediated positive feedback is not a necessary prerequisite for synchronous sister chromatid separation and that instead the separation dynamics are explainable purely from stochastic separase-mediated cohesin cleavage. Fitting a stochastic model of cohesin cleavage to our experimental results suggests that small-number effects in cohesin at the time of sister chromatid separation lead to inevitable variability in segregation timing, i.e., synchrony is naturally limited by molecular noise.

## Results

### Sister chromatids separate within a narrow time window—but not with perfect synchrony

In fission yeast, securin (Cut2) degradation is completed in about 4 min, and chromosome I splits about 2 min after the onset of securin degradation (Fig. S1, A–D) (Kamenz and Hauf, 2014). To analyze the time window in which the three chromosomes separate, we used live-cell imaging with high temporal resolution (3.5–7 s) and monitored fluorescent fusion proteins recruited to the centromeres of the three fission yeast chromosomes (Fig. 1, A and B). On chromosome II, LacI-GFP was targeted to a position close to the centromere with *lacO* repeats (cen2-GFP); on chromosomes I and III, TetR-tdTomato was targeted to a position close to the centromere with *tetO* repeats (centromere 1 [cen1]- and cen3-tdTomato) (Sakuno et al., 2009; Straight et al., 1996; Yamamoto and Hiraoka, 2003). We compared separation timing between chromosomes I and II or II and III and defined one chromosome as separating when sister chromatids started a sustained

movement toward opposite spindle poles and ceased to move coordinately or convergently (Fig. 1, C and D). We typically scored separation based on visual inspection of the time-lapse recording, but tracking of the separation trajectories confirmed that the visually assigned separation times align with the onset of rapidly increasing inter-centromere distance (Fig. S1 E).

The time difference of separation between two chromosomes (Δt, Fig. 1 D) reflects the synchrony of anaphase; a 0 s time difference indicates perfect synchrony. Indeed, monitoring two centromere markers on the same chromosome showed highly synchronous separation with a mean Δt of 0.6 s (Fig. 1 E and Fig. S1 F). However, slight differences in separation timing between the markers were possible (standard deviation 6 s), suggesting that separation of different parts of the centromere has a stochastic component. With markers on two different chromosomes, the mean of Δt was still close to, but distinct from, zero, indicating a bias in the centromere of one chromosome separating before the other. cen1 split on average 3.4 s before cen2, and cen2 split on average 11.5 s before cen3 (Fig. 1 F). However, this order was not absolute; despite cen2 separating on average 11.5 s before cen3, cen3 separated before cen2 in about 23% of the cells. The variations in Δt between individual cells, measured as the standard deviation of Δt, were 19 s for cen1 versus cen2 and 14 s for cen2 versus cen3, broader than for both markers on the same chromosome (6 s) (Fig. 1, E and F; and Table S1), indicating additional stochastic differences between the two chromosomes. The type of growth medium (rich versus minimal) did not have a significant influence on separation timing (Fig. S1 G).

In budding yeast, it was initially proposed that chromosome segregation occurs in a strictly sequential order (Holt et al., 2008), but this was later attributed to one of the fluorescent tags (*tetO*/TetR-GFP) creating abnormally strong cohesion and delaying the separation of one chromosome (Lyons and Morgan, 2011; Fuchs et al., 2002). We therefore swapped the *tetO* and *lacO* markers for cen1 and cen2. While the presence of the *tetO* array may indeed very slightly bias separation toward later times, we did not observe any reversal in separation order, and the overall distribution remained similar (Fig. S1 H), indicating that separation timing is not greatly influenced by the nature of the fluorescent tag in our system.

The three centromere tags have different distances from the central core of the centromere where the kinetochore assembles and microtubules attach, and the marker on chromosome III was the furthest away from the central core (Fig. 1 B). It was therefore possible that different separation timings reflect a "peeling apart" of the centromere region until the marker is reached. However, placing tags at the central core of the centromere, rather than the periphery, still resulted in similar Δt distributions (Fig. 1, G and H). The bias for the centromere of chromosome II separating before that of chromosome III was slightly reduced but remained present (7.8 vs. 11.5 s). This suggests that the centromeric region may separate as a single coherent unit. The chromosome arms, in contrast, separate distinctly later. A marker at the end of the long arm of chromosome II (Ding et al., 2004) split between 30 s and >3 min after the chromosome II centromere (Fig. S1, I–K). This is consistent with observations of the arms progressively peeling

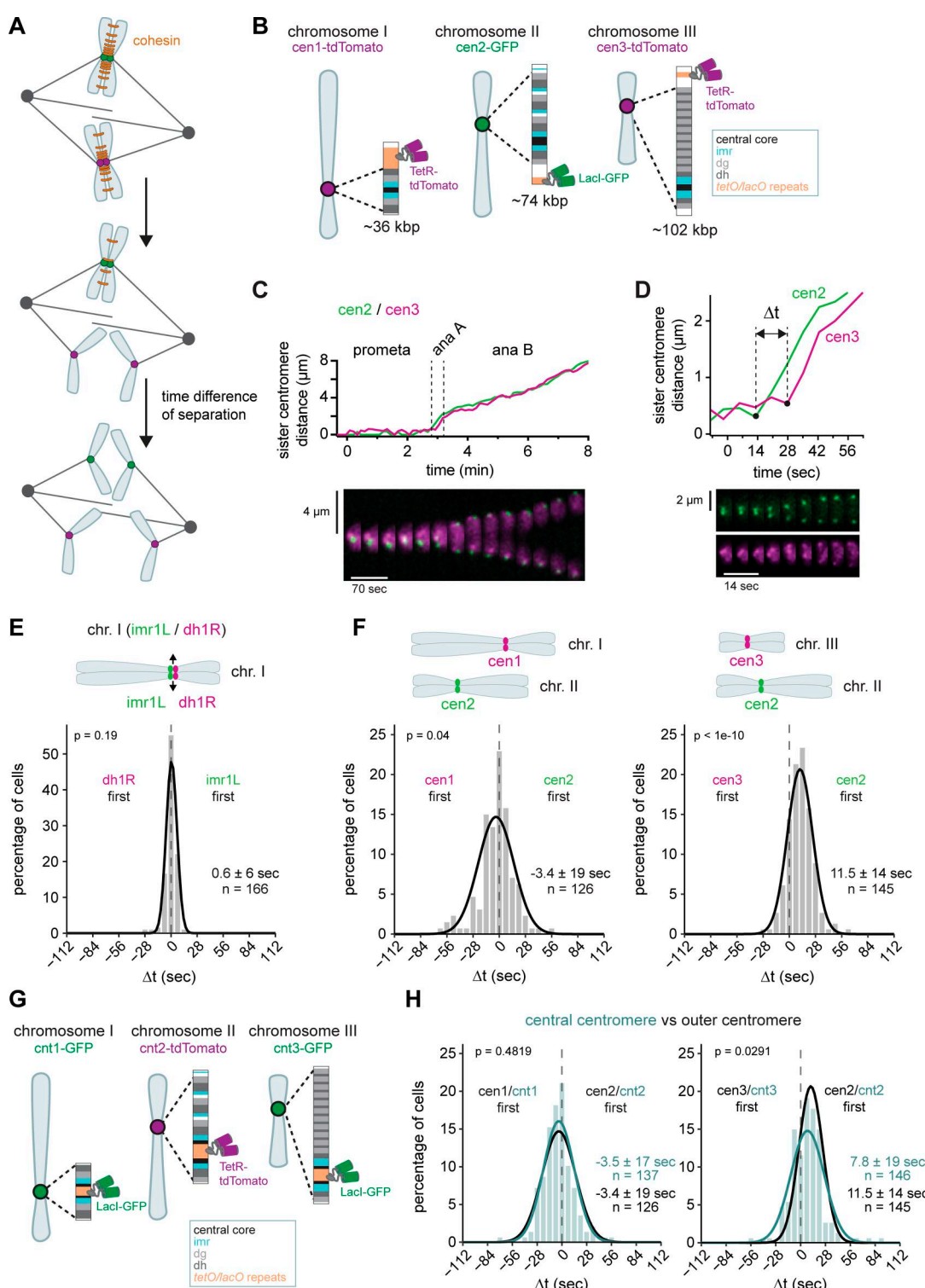

Figure 1. **Centromeres segregate within a narrow time window but not with perfect synchrony. (A)** Schematic depicting cohesin cleavage and subsequent sister chromatid separation; cohesin complexes in orange. **(B)** Fluorescent labeling of chromosomes close to their centromeric regions. The schematic depicts the localization of the tandem *tetO* or *lacO* repeats relative to the centromere, which comprises central core (cnt), innermost repeats (imr), and different numbers of dg/dh repeat pairs. Chromosome II was marked with *lacO*/LacI-GFP (Yamamoto and Hiraoka, 2003); chromosome I and III were marked with *tetO*/TetR-tdTomato (Sakuno et al., 2009 and this study). The lengths of the centromere regions (distance between furthest dh/dg repeats) are shown in kbp. **(C and D)** Sister centromere distance and corresponding kymograph for a strain with cen2-GFP and cen3-tdTomato markers. The same cell in C and D. **(D)** shows how the time difference between two markers (Δt) is scored. **(E and F)** Frequency distributions and Gaussian fit (continuous lines) of the time difference (Δt) between the separation of two markers, either on the same chromosome (E) or on chromosomes I and II or chromosomes II and III (F). Mean ± SD of the fit and number of cells are shown; P value from the one-sample *t* test against zero. **(G)** Fluorescent labeling of chromosomes at the inner centromeric regions. Chromosome II was marked with *tetO*/tetR-tdTomato (Sakuno et al., 2009); chromosomes I and III were marked with *lacO*/LacI-GFP (Sakuno et al., 2009 and

this study). **(H)** Cyan: Frequency distributions and Gaussian fit (continuous lines) of the time difference (Δt) between the separation of central centromere markers on chromosomes I and II or chromosomes II and III. The fitted Gaussian distributions for cells using outer centromere markers (F) are shown in black for comparison. Mean ± SD of the fit; *n* = number of cells; P values from a two-sample Kolmogorov–Smirnov test.

apart in both fission yeast and other eukaryotes (Ding et al., 2004; Paliulis and Nicklas, 2004; Renshaw et al., 2010; Chu et al., 2022). For the remainder of the experiments, we focused on centromere segregation as the earliest event of chromosome splitting.

Overall, these data quantify the synchrony of anaphase (∼15–20 s SD between two chromosomes) and indicate that centromere separation has a considerable stochastic component so that the order of chromosome separation varies between cells.

### Chromosome segregation synchrony depends more on separase activity than on microtubule dynamics

The separation of sister chromatids is triggered by separase-mediated cohesin cleavage (Uhlmann et al., 2000); as such, synchrony likely requires rapid cohesin cleavage, which, in turn, is controlled by separase activation. Reducing separase activity, therefore, should make chromosome separation more asynchronous (reflected in a wider spread of Δt). To test this, we used the temperature-sensitive separase mutant *cut1-206* (Hirano et al., 1986) and followed sister chromatid separation at a semi-permissive temperature. When impairing separase activity, centromere separation for both chromosome pairs became significantly more asynchronous, more than doubling the standard deviation (Fig. 2 A, Fig. S2, A–D, and Table S1). This is consistent with observations in budding yeast, where a separase mutant slowed down cohesin cleavage (Yaakov et al., 2012) and increased the variation in separation time between two chromosomes (Holt et al., 2008; Lyons and Morgan, 2011). We further observed a slower movement of centromeres towards the spindle poles in the separase mutant (Fig. 2 B and Fig. S2 E). This indicates a sustained requirement for separase activity beyond centromere splitting. Consistently, in budding yeast, the time between centromere separation and arm separation became longer when separase activity was reduced (Renshaw et al., 2010), which has been attributed to a requirement to remove remaining chromosome arm cohesin during early anaphase.

To address whether separase activity is limiting for chromosome separation synchrony in WT cells, we overexpressed separase. We co-overexpressed the inhibitor securin (Fig. S3, A and B), both to prevent the potential lethality associated with uninhibited separase (Kamenz and Hauf, 2014) and to accelerate separase release since securin overexpression accelerates securin degradation kinetics (Kamenz et al., 2015). Neither securin overexpression alone (Fig. S3 C) nor securin and separase co-overexpression (Fig. 2 C and Fig. S3 D) significantly reduced the variation of Δt. This suggests that, even though separase activity is important for synchronous anaphase (Fig. 2 A), additional factors limit anaphase synchrony. Sister chromatids still moved with the same speed in anaphase A in cells co-overexpressing securin and separase (Fig. 2 D; and Fig. S3, E and F), suggesting

that cleavage of remaining cohesin complexes on arms is not limited by Sep availability.

Since these experiments suggested that mechanisms other than separase activity limit anaphase synchrony, we sought to determine whether forces exerted by microtubules of the mitotic spindle play a role. Microtubules undergo growth and shrinkage, and kinetochores attached to microtubules oscillate between the spindle poles, partially also influenced by motor proteins (VandenBeldt et al., 2006; Cheeseman and Desai, 2008; Walczak et al., 2010). Hence, the separation of centromeres may be delayed after cohesin cleavage until microtubules pull toward opposing poles. Indeed, in *Drosophila*, microtubule flux has been shown to contribute to anaphase synchrony (Matos et al., 2009). We used two perturbations to impair microtubule dynamics: the microtubule drug MBC and deletion of the kinesin-8, *klp5*. MBC is expected to weaken microtubules, and *klp5Δ* is expected to stabilize microtubules and alter chromosome oscillations (Balestra and Jimenez, 2008; Garcia et al., 2002; West et al., 2002; Unsworth et al., 2008; Klemm et al., 2018). We titrated MBC to a concentration that still allowed proper chromosome segregation in almost all cells but slowed down anaphase centromere movement (Fig. 2 F; and Fig. S4, A and B). Anaphase synchrony was slightly worsened by MBC—the standard deviation of Δt for cen1 vs. cen2 increased from 19 to 22 s, and for cen2 vs. cen3 from 14 to 22 s (P = 0.3970 and 1.92e−4, respectively, by Levene's test) (Fig. 2 E). However, this change was considerably less than that observed in the Sep mutant, which—while having similarly slow centromere movement during anaphase (Fig. 2 B)—had standard deviations of 35 s for cen1 vs. cen2 and 34 s for cen2 vs. cen3 (P = 4.78e−9 and 1.17e−7, respectively, by Levene's test) (Fig. 2 A). Furthermore, anaphase synchrony was unchanged in *klp5Δ* cells (Fig. S4 C), even though chromosomes were often misaligned at the onset of anaphase, as expected (Fig. S4 D). Together, our data therefore suggest that microtubules play a minor role in *Schizosaccharomyces pombe* centromere separation synchrony compared with separase.

### No evidence for separase-mediated feedback on securin degradation to enhance chromosome segregation synchrony

As separase activity plays an important role in the synchrony of centromere separation, we considered possible mechanisms that allow for a rapid, sudden increase in separase activity. In budding yeast, a positive feedback loop has been proposed to contribute to synchrony, where separase enhances the degradation of its inhibitor, securin, by promoting the dephosphorylation of CDK1-dependent phosphorylation sites on securin (Holt et al., 2008). Persistent CDK1 activity (through expression of a nondegradable version of the cyclin Clb5) decreased sister chromatid separation synchrony. However, it is unclear whether this is caused by disruption of the proposed feedback loop or by the sustained CDK1 activity, independent of securin degradation.

We conducted a similar experiment in fission yeast and conditionally expressed a nondegradable version of the B type

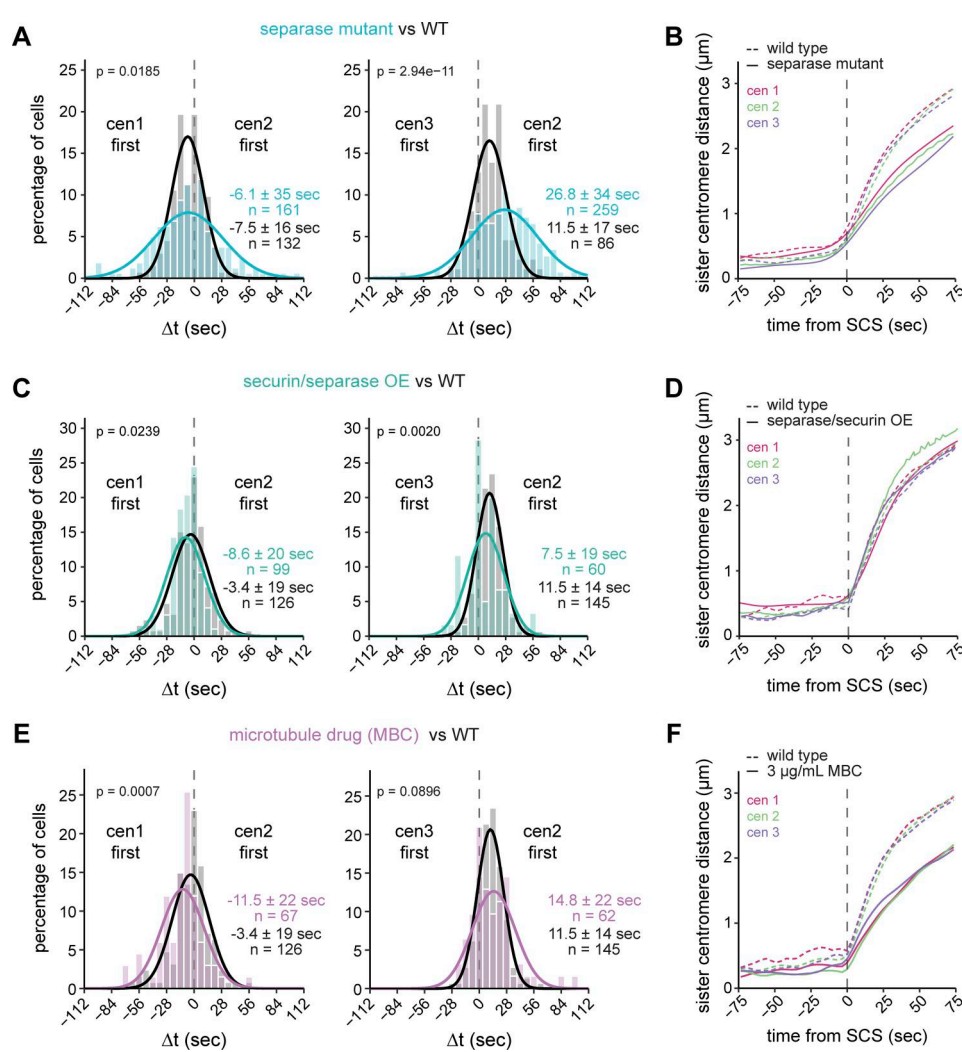

Figure 2. **Separase activity has a larger influence on chromosome segregation synchrony than microtubule dynamics. (A, C, and E)** Frequency distributions and Gaussian fit (continuous lines) of the time difference between the separation of centromeres 1 and 2 or centromeres 2 and 3 for cells carrying the temperature-sensitive separase allele *cut1-206* (A), cells with overexpressed securin and separase (C), and cells exposed to 3 µg/ml of the microtubule drug MBC (E). The fitted Gaussian distributions of WT cells (black), imaged in the same medium as the experimental strain, are shown for comparison (*cut1-206*: rich medium; securin/separase overexpression and MBC treatment: minimal medium). Mean ± SD of the fit; *n* = number of cells; P values from a two-sample Kolmogorov–Smirnov test. See Figs. S2, S3, and S4 for additional data. **(B, D, and F)** Distances between sister chromatid pairs in WT cells (dashed lines) and cells carrying the temperature-sensitive separase allele *cut1-206* (B, solid lines), cells with separase and securin co-overexpression (D, solid lines), or cells exposed to 3 µg/ml MBC (F, solid lines). Colors correspond to the chromosome identity. Distances are aligned to the sister chromatid separation (SCS) time of the respective chromosome. At least 17 cells were averaged for each chromosome and genotype; exact cell counts are shown in Figs. S2, S3, and S4.

cyclin Cdc13 (ΔN-cyclin B, lacking residues 1–67) at close to endogenous levels (Kamenz and Hauf, 2014). As a consequence, a fraction of cells was unable to divide the nucleus and exit mitosis, as is expected from sustained CDK1 activity (Yamano et al., 1996). Other cells, presumably with lower levels of nondegradable cyclin B, still underwent nuclear division and exited mitosis (Fig. 3, A and B). The synchrony of sister chromatid separation in both classes of cells was comparable with that of WT cells (Fig. 3 A and Table S1). This made it unlikely that fission yeast has a similar type of feedback loop as proposed for budding yeast. Consistent with the similar separation synchrony, the securin degradation rate also remained highly similar to that in WT cells when nondegradable Cdc13 was expressed (Fig. S4 E) (Kamenz and Hauf, 2014), making it unlikely that separase affects securin

degradation through CDK1. To test more broadly whether separase enhances securin degradation—independent of CDK1—we expressed two versions of securin: WT securin tagged with GFP to monitor securin degradation and inducible, untagged, nondegradable securin (ΔN-securin, lacking residues 1–75) to block separase activity. If separase enhances securin degradation, securin degradation should be slowed down when separase release is blocked by nondegradable securin. We confirmed this prediction by implementing a computational model for the key reactions with or without feedback (Fig. 3, C and D; see Materials and methods). Experimentally, we found that the presence of nondegradable securin blocked nuclear division (Fig. 3 F), indicating a failure of separase activity; yet, degradation of the WT version of securin was unaffected (Fig. 3, E and F; and Fig. S4 F).

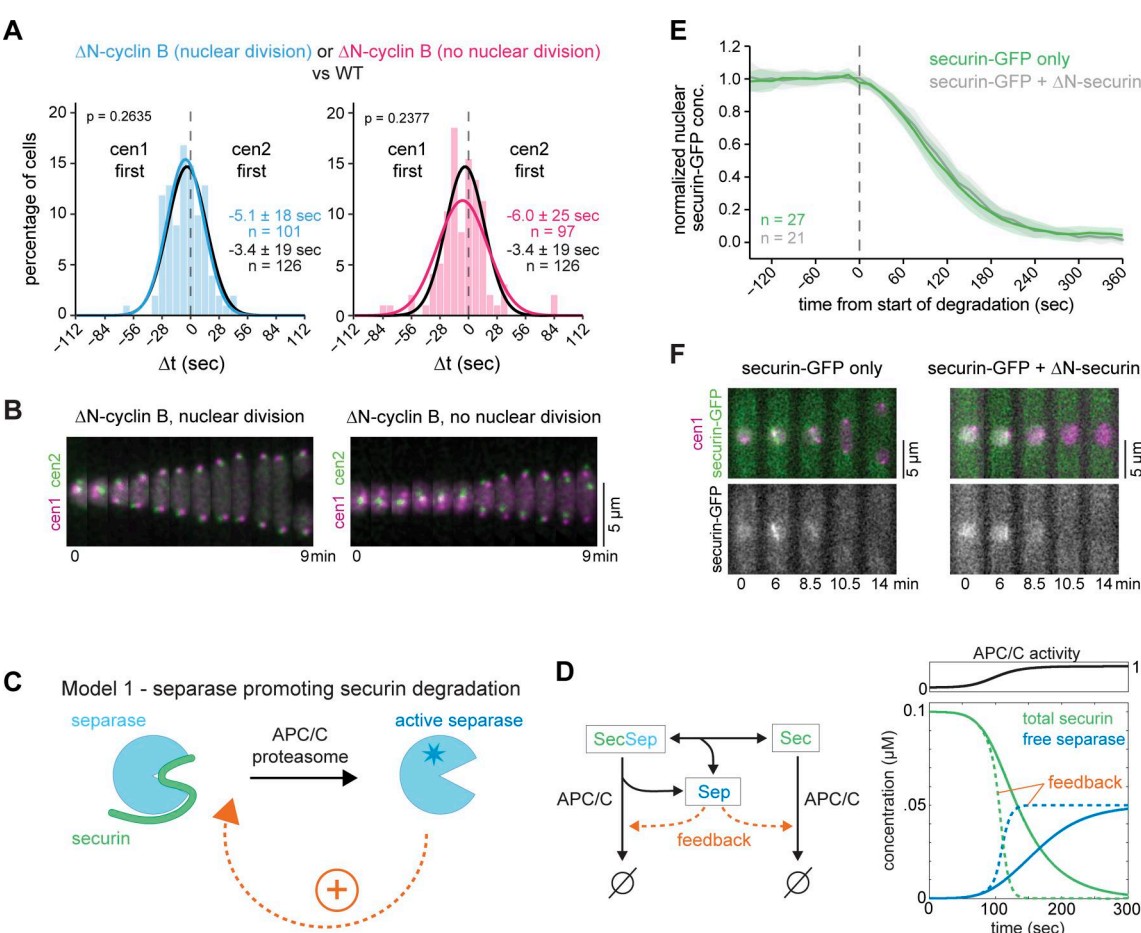

Figure 3. **Separase-mediated feedback on securin degradation is unlikely. (A)** Frequency distribution and Gaussian fit (continuous lines) of the time difference between the separation of centromeres 1 and 2 after expression of nondegradable cyclin B (Cdc13), ΔN-cyclin B, in minimal medium. The fitted Gaussian distributions of WT cells grown under similar conditions (black) are shown for comparison. Mean ± SD of the fit; *n* = number of cells; P values from a two-sample Kolmogorov–Smirnov test. **(B)** Representative kymographs of sister chromatid separation in cells with cen1-tdTomato and cen2-GFP after expression of nondegradable cyclin B (Cdc13), ΔN-cyclin B. The left panel displays a cell that undergoes nuclear division and progresses through mitosis; the right panel displays a cell that fails to undergo nuclear division. **(C)** Schematic depicting a positive feedback loop, where separase accelerates its own activation by accelerating securin degradation. **(D)** Computational model for APC/C-mediated securin (Sec) degradation and separase (Sep) release with or without feedback of separase on securin degradation. Left side: diagram of the reactions in the model. Right side: simulation with or without feedback (dashed and solid lines, respectively) assuming a sigmoidal increase of APC/C activity with time. See Materials and methods for model details. **(E)** Securin-GFP degradation in WT cells (green) or in cells after induction of nondegradable securin (Cut2), ΔN-securin (gray). Individual time courses are normalized and aligned to the start of securin degradation at t = 0. Mean (line) ± SD (shaded area) of the cell population; *n* = number of cells. **(F)** Representative kymographs of securin-GFP degradation in a WT cell (left panel) and in a cell after induction of untagged nondegradable securin (Cut2), ΔN-securin. In the latter cell, the nucleus fails to divide.

Hence, separase activity does not accelerate securin degradation in fission yeast. We conclude that a positive feedback loop, where separase enhances its own activation by accelerating securin degradation, is not an integral part of the mechanism that promotes synchronous sister centromere separation in fission yeast.

**No evidence for separase autoactivation to enhance chromosome segregation synchrony**

A positive feedback loop would not necessarily need to act upstream to promote separase release from securin (Fig. 3 C); it could also operate downstream, such as through separase autoactivation (Fig. 4 A). For example, metazoan separase is known to cleave itself autocatalytically, which could alter its activity (Waizenegger et al., 2002; Zou et al., 2002; Chestukhin et al.,

2003; Shindo et al., 2022). However, there is no clear evidence that cleavage enhances activity (Papi et al., 2005; Holland et al., 2007; Shindo et al., 2022), and separase auto-cleavage has not been observed in yeast (Hornig et al., 2002). Nevertheless, to analyze the possibility of downstream positive feedback, regardless of the mechanism, we simulated securin degradation and separase release with or without separase autoactivation in a computational model (Fig. 4, A and B; see Materials and methods). The two model versions differ in their response to decreased APC/C activity. With or without feedback, decreased APC/C activity slows down securin degradation; however, without feedback, separase release and the increase of active separase become more gradual (Fig. 4 B, left graph; Fig. S4 G), whereas with feedback, the increase in separase activity remains abrupt but occurs later (Fig. 4 B,

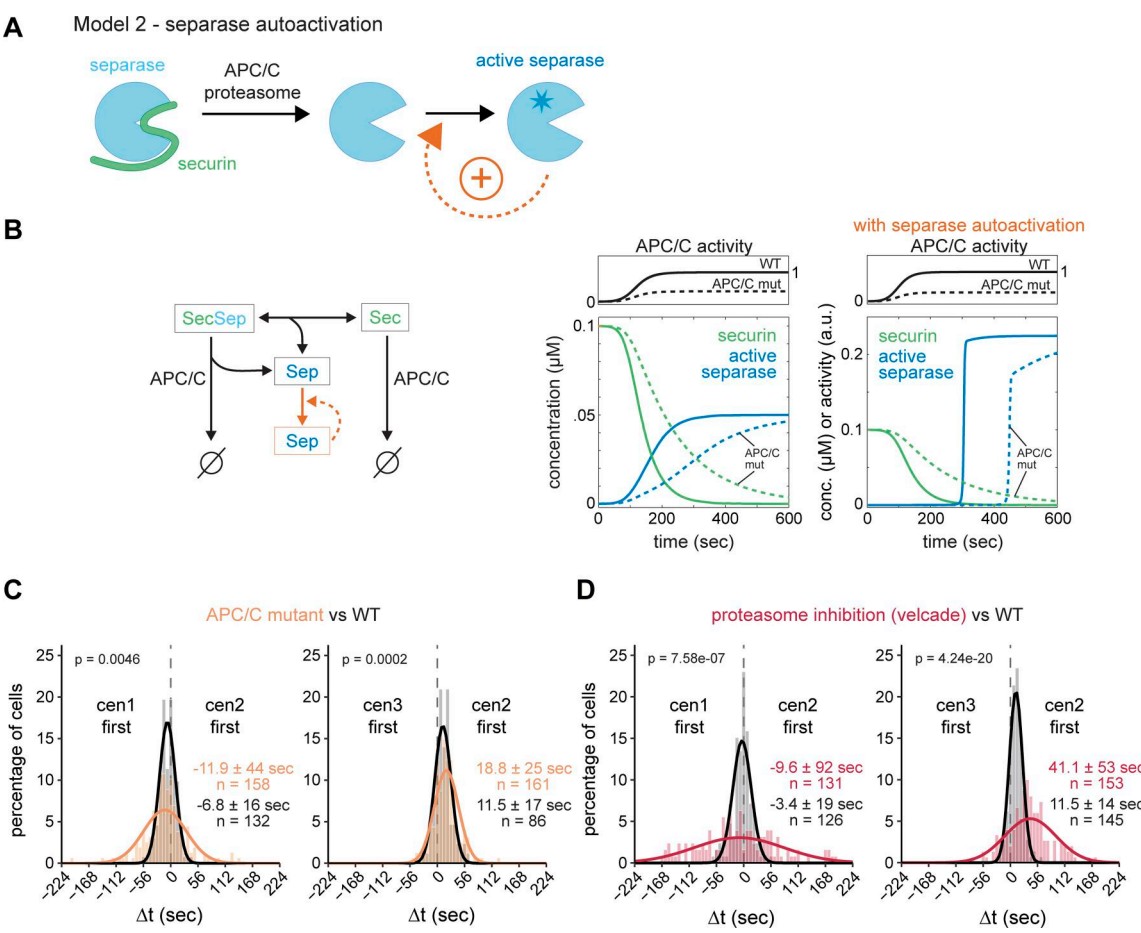

Figure 4. **Separase autoactivation is unlikely. (A)** Schematic depicting a positive feedback loop, where separase accelerates its own activation downstream of securin degradation. **(B)** Computational model for separase (Sep) release mediated by APC/C-mediated securin (Sec) degradation with or without separase autoactivation. Left side: diagram of the reactions in the model. Right side: simulation for high or low APC/C activity, without or with separase autoactivation, assuming a sigmoidal increase of APC/C activity with time. Solid lines indicate high APC/C activity; dashed lines indicate low APC/C activity, mimicking an APC/C mutant. See Materials and methods for model details. **(C and D)** Frequency distributions and Gaussian fit (continuous lines) of the time difference between the separation of centromeres 1 and 2 or centromeres 2 and 3. Cells were carrying either a temperature-sensitive allele of the APC/C subunit Cut9 (*cut9-665*) and were grown in rich medium before imaging (C, orange) or were grown in minimal medium and treated with 100 µM of the proteasome inhibitor velcade (bortezomib) 30 min prior to imaging (D, red). The fitted Gaussian distributions of WT cells grown under similar conditions and without inhibitor are shown for comparison (black). Mean ± SD of the fit; *n* = number of cells; P values from a two-sample Kolmogorov–Smirnov test.

right graph; Fig. S4 G). If separase autoactivates, one would therefore expect that reduced APC/C activity delays sister chromatid separation but does not affect synchrony.

To test this, we used cells with a temperature-sensitive allele of the APC/C subunit APC6 (*cut9-665*) grown at a semi-permissive temperature (Hirano et al., 1986). Securin degrades slower in this mutant, as expected (Fig. S4, H and J). Inconsistent with separase autoactivation, sister chromatid separation became less synchronous (standard deviations of Δt 44 and 25 s) (Fig. 4 C and Table S1). To corroborate this result, we slowed down securin degradation with a second method—by partial inhibition of the proteasome with Velcade (bortezomib) (Takeda et al., 2011). This yielded even slower securin degradation than in the APC/C mutant (Fig. S4, I and J) and further decreased the synchrony of sister chromatid separation (standard deviations of Δt 92 and 53 s) (Fig. 4 D, Fig. S4 K, and Table S1). Because of this pronounced effect of securin degradation kinetics on the synchrony of sister chromatid separation, separase autoactivation is unlikely.

**The dynamics of chromosome segregation can be reproduced by a basic stochastic model of cohesin cleavage**

Since our results argue against separase-mediated positive feedback, we sought to determine if minimal features of separase regulation are sufficient to account for the sister chromatid separation dynamics in WT and mutant cells. To test this, we implemented a stochastic model of separase-mediated cohesin cleavage (Fig. 5 A; see Materials and methods). We assumed that separase activity increases gradually over the period τ until it reaches its maximal rate ($k_{max}$) and that separase stochastically cleaves cohesin complexes on the three different chromosomes. Sister chromatid separation occurs when the initial number of cohesin complexes on one chromosome (*N*) has been reduced to the threshold number *n*. We do not necessarily assume that the threshold number *n* is zero, because, if the forces on a cohesin complex are large enough, it will let go of the bound DNA even without cleavage (Daum et al., 2011; Richeldi et al., 2024). We set boundaries for all parameters based on our data and the

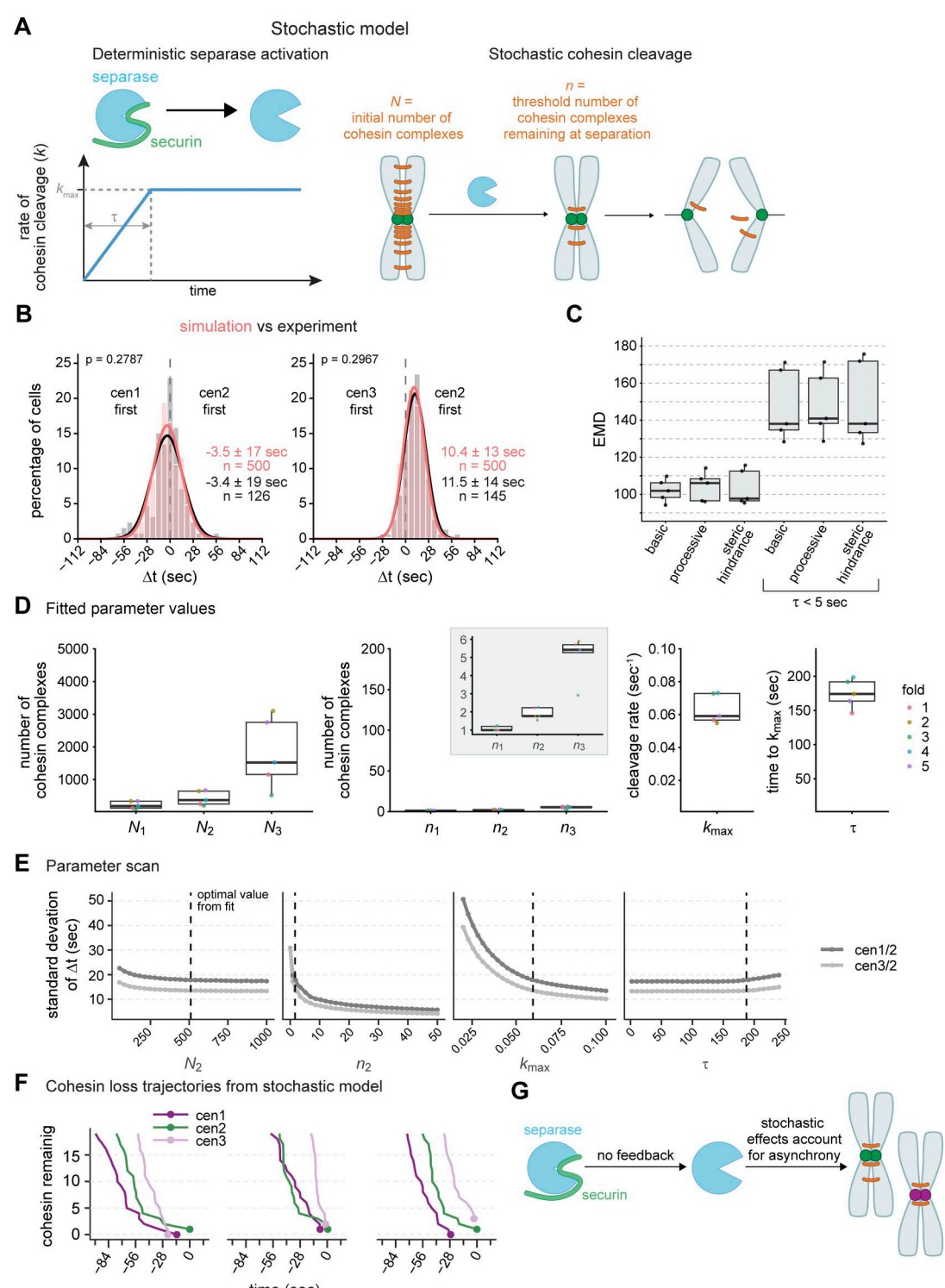

Figure 5. **Small-number effects cause stochasticity in sister chromatid separation time. (A)** Schematic of the stochastic model of separase-mediated cohesin removal. Separase activation (through securin degradation) is represented by a constant increase in the cohesin cleavage rate, $k(t)$. The rate increases for a period $\tau$ until it reaches its maximum value, $k_{max}$. Each chromosome initially has $N$ cohesin complexes ($N_1$, $N_2$, and $N_3$ for chromosomes I, II, and III, respectively), which are randomly cleaved by separase. Chromosomes separate once the number of remaining cohesins falls below a threshold $n$ ($n_1$, $n_2$, and $n_3$ for chromosome I, II, and III, respectively). See Materials and methods for model details. **(B)** Frequency distributions and Gaussian fit (continuous lines) of the time differences between the separation of centromeres 1 and 2 or centromeres 2 and 3, either from a simulation using optimal parameter values of the basic model (salmon) or determined experimentally in wild-type cells (gray/black, same data as in Fig. 1 F). Mean ± SD of the simulation results or the Gaussian fit (for experimental data); $n$ = number of cells; P values from EMD. **(C)** Goodness of fit of the model variants measured by EMD (see Materials and methods) with and without constraining $\tau$ at low values ($\tau < 5$ s). Points indicate the validation EMD values obtained in the fivefold cross-validation (see Materials and methods); boxplots show median, interquartile range, and range. Basic, basic model with time-dependent cleavage rate alone; processive, model with bursts of cohesin

cleavage; steric hindrance, model in which only a surface fraction of cohesin is available for cleavage and inner layers of cohesin are progressively exposed (see Materials and methods). **(D)** Parameter values of the basic model fitted to the experimental data. Points indicate values obtained in each of the fivefolds (see Materials and methods); boxplots show median, interquartile range, and range; the y-axis spans the parameter range tested, except for the inset showing $n_1$, $n_2$, and $n_3$. **(E)** One-at-a-time (OAT) sensitivity analysis of the basic model. Each parameter was varied individually across its allowed range while all other parameters were held fixed at their optimal values obtained by fitting. 10 batches of 10,000 simulations were performed for each parameter point. Each batch yielded a mean and SD of the separation time differences ($\Delta t$) between cen1 and centromere 2, and centromere 2 and 3. Plot shows mean of the $\Delta t$ SD ± SEM across 10 batches. Dashed vertical lines indicate the optimal parameter values. **(F)** Stochastic cohesin loss trajectories near the time of separation in the basic model. Simulated cohesin counts for chromosomes 1, 2, and 3 are shown for three representative simulations, aligned to t = 0 at the final separation event. Lines indicate cohesin loss over time; points mark the moment of separation for each chromosome. **(G)** Schematic summarizing the results: separase becomes active upon securin degradation without positive feedback. If a small number of cohesin complexes suffice to maintain cohesion, stochasticity of the final cleavage events dominates the timing of sister chromatid separation, leading to asynchrony and no clear order between chromosomes.

available literature (Table S4). The parameters we estimate are: $\tau$, $k_{max}$, the initial number of cohesin molecules on each chromosome ($N_1$, $N_2$, and $N_3$), and the remaining number of cohesin molecules on each chromosome at the time of separation ($n_1$, $n_2$, and $n_3$). We fitted the model to the experimental data from WT cells, as well as all perturbations (separase and APC/C mutants, velcade, and MBC) (see Materials and methods). We assumed (1) that the separase mutant changes separase activity, which we represent as a reduction in $k_{max}$; (2) that the APC/C mutant and velcade—by slowing down securin degradation—change the speed of separase activation, which we represent as a longer $\tau$; and (3) that MBC—by lowering microtubule forces—reduces how many cohesin molecules can remain at the time of separation, which we represent as a lower threshold number, $n$. Goodness of fit between the model and data is assessed by the Earth Mover's Distance (EMD).

This model was able to reproduce the trends in sister chromatid separation synchrony across all experimental conditions (Fig. 5 B and Fig. S5 A), providing additional evidence that feedback regulation is not necessary. In fact, forcing $\tau$ to be very short, which mimics the separase activity dynamics with separase autoactivation (Fig. 4 B), makes the fit worse (Fig. 5 C, basic vs. basic $\tau < 5$ s; Fig. S5 A). To further probe whether adding regulatory mechanisms to the basic model would improve the fit, we considered two plausible variations: (1) processive separase action, where not one but multiple cohesin molecules are removed by each separase action, and (2) steric hindrance, where initially not all cohesin is available to separase and only becomes available progressively (see Materials and methods). While both these additions provide good fits, they do not substantially improve the fit (Fig. 5 C). Hence, a basic stochastic model of separase-mediated cohesin cleavage is sufficient to explain the experimentally observed dynamics of sister chromatid separation. In contrast, assuming separase autoactivation ($\tau < 5$ s) produces dynamics that are significantly less consistent with the experimental data (Fig. 5 C and Fig. S5 A).

**Small-number effects limit separation synchrony**
To identify the main determinants of separation synchrony, we inspected the values of the parameters obtained by fitting (Fig. 5 D) as well as the model's sensitivity to changes in individual parameters (Fig. 5 E; and Fig. S5, B and C). The optimal value for the starting number of cohesin molecules on each chromosome ($N$) was highest on chromosome 3, lower on chromosome 2, and yet lower on chromosome 1 (Fig. 5 D). This is explained by the

fact that the ratio of $N$ between two chromosomes influences the mean $\Delta t$ (Fig. S5 B). Hence, the different starting numbers help account for the preferential order observed (chromosome 1 being slightly more likely to separate before chromosome 2, and chromosome 2 more often separating before chromosome 3 than *vice versa*, Fig. 1 F and Fig. S1 G).

Two parameters, $n$ and $k_{max}$, strongly influence the synchrony of sister chromatid separation (Fig. 5 E). Most notably, the value of $n$ (threshold number of cohesin molecules remaining at the time of separation) that we obtained by fitting was at the lowest edge of the allowed parameter range (Fig. 5 D). In this low-number regime, the timing of the final few cohesin cleavage events is stochastic, which decreases synchrony in the separation time and randomizes the order of separation (Fig. 5, E and F; and Fig. S5 C). Synchrony is also decreased by low separase activity (Fig. 5 E) because the timespan between stochastic degradation events and, therefore, between final separation events increases. Together, these results indicate that small-number effects place a natural limit on the synchrony of sister chromatid separation. In other words, if a small number of cohesin molecules is sufficient for cohesion (which is plausible, see Discussion), asynchrony in chromosome separation is expected.

## Discussion
Anaphase is visually one of the most striking steps of the cell cycle. When mitosis is imaged in live cells, the splitting of chromosomes into their sister chromatids occupies only a short period within mitosis and appears abrupt and synchronous. However, imaging at high temporal resolution, as we did here using fission yeast and as has been done in human cells (Armond et al., 2019, *Preprint*; Sen et al., 2021; Harrison et al., 2021, *Preprint*), reveals that sister chromatid separation is not perfectly synchronous and that there is significant variation in the segregation time of single chromosomes (Fig. 1). We propose that there is a natural limit to separation synchrony resulting from small-number effects.

**Separase-mediated positive feedback is not necessary to explain the dynamics of sister chromatid separation**
In budding yeast, lower separase activity makes chromosome segregation more asynchronous (Holt et al., 2008; Lyons and Morgan, 2011) (Fig. 2), and a focus of prior studies has therefore been on separase regulation. Positive feedback loops are common in the regulation of cell cycle transitions and support

both a rapid transition and irreversibility (Kapuy et al., 2009; Ferrell and Ha, 2014; Ferrell, 2013). It was therefore reasonable to assume that anaphase, as one of the major cell cycle transitions, makes use of positive feedback (Holt et al., 2008). Our results, however, argue against separase activity-enhancing positive feedback in anaphase regulation in fission yeast (Figs. 3 and 4). Anaphase is special among the cell cycle transitions, as its irreversibility is inherent in its mechanics—the loss of chromosome cohesion cannot be reversed. Sister chromatid separation in anaphase may therefore be a major cell cycle transition that can dispense with positive feedback.

Other non-feedback mechanisms support rapid separase activation and cohesin cleavage, though (Kamenz and Hauf, 2017). Securin is a stoichiometric inhibitor of separase, present in excess over separase (Ciosk et al., 1998; Shindo et al., 2012; Hellmuth et al., 2014; Kamenz et al., 2015). Sequestration of separase by securin allows, in principle, for a rapid activation of separase, once the excess securin pool is exhausted—known as inhibitor ultrasensitivity (Buchler and Cross, 2009; Ferrell and Ha, 2014; Legewie et al., 2008; Hopkins et al., 2017). In human and mouse cells, unbound securin is degraded before separase-bound securin (Hellmuth et al., 2014; Thomas et al., 2021), reinforcing that the free pool is exhausted first. Various posttranslational modifications on securin, separase, and cohesin also support separase activity and separase-mediated cohesin cleavage (Alexandru et al., 2001; Yaakov et al., 2012; Li et al., 2017; Wang et al., 2024; Lianga et al., 2018; Yu et al., 2025). Even with such regulation in place, though, separase release will still slow down when securin degradation is slowed, consistent with our experimental results (Fig. 4). However, when we raised separase activity (Fig. 2), sister chromatid separation remained asynchronous. This suggests that once separase activation is sufficiently fast, increasing its activity no longer strongly improves synchrony. We propose a limit to separation synchrony that is determined by the nature of cohesin-mediated cohesion.

### Small-number effects as a natural limit to chromosome separation synchrony

Fitting a stochastic model of separase-mediated cohesin cleavage to our data revealed that the observed variation in chromosome separation order can be explained by the stochasticity that arises when a small number of cohesin molecules suffices for cohesion (Fig. 5). Is this plausible? Maybe yes. Studies that titrated cellular cohesin levels found that sister chromatid cohesion was maintained at levels as low as 13% of cohesin in budding yeast (with lower levels not tested) or 22% in *Drosophila* cells (Heidinger-Pauli et al., 2010; Carvalhal et al., 2018). More directly, the force required to rupture a fission yeast cohesin complex connecting two DNA molecules has been carefully measured *in vitro* (Richeldi et al., 2024) and was found to be in the same range as the force that a kinetochore-microtubule is thought to generate (Gudimchuk and Alexandrova, 2023; Nicklas, 1983; Akiyoshi et al., 2010; Volkov et al., 2013). Yeast kinetochores attach to 1–4 microtubules (Ding et al., 1993; Winey et al., 1995; O'Toole et al., 1999), suggesting that indeed a few cohesin molecules may suffice to maintain cohesion, consistent with the expectations from our stochastic model.

In metazoan cells, with a larger number of kinetochore microtubules, more cohesin complexes will be required to maintain cohesion (possibly in the order of ~40 [Richeldi et al., 2024]). This may reduce the stochasticity in chromosome separation and establish a more predictable order determined by the difference in the initial amount of cohesin at the centromeres. Short-time-frame imaging in human cells found that separation of the first and last chromosome can be spaced by as much as 90 s (Armond et al., 2019, *Preprint*; Sen et al., 2021). However, individual chromosomes were not identified, and it therefore remains unclear how much this reflects a defined temporal order versus stochastic variation in separation times. Chromosome spreads of early anaphase chromosomes from metazoan cells suggest that chromosomes with smaller centromeres separate before those with larger centromeres (Vig, 1983). In addition, a distinct order in the separation of different mammalian chromosomes was also inferred from an observed inheritance of chromosome positions within the nucleus (Gerlich et al., 2003). Together, this suggests that the order of chromosome separation may be more pronounced in mammalian cells compared with fission yeast, possibly because small-number effects are minimized.

In summary, we propose that asynchrony in fission yeast sister chromatid separation, and potentially other cell types, arises because only a handful of cohesin molecules are left when cleaving one more leads to separation (Fig. 5). We base this on the good fit between our experimental data and a basic stochastic model of cohesin cleavage. In the spirit of Occam's razor, using a simple model seems justified, but we acknowledge that stochastic variations in separation order could also result from factors not covered in our model. This includes stochastic variations in the amount of cohesin loaded onto single chromosomes, variation in centromere packing that alters access to cohesin, or variation in spindle forces. In *Drosophila* cells, for example, microtubule flux homogenizes the microtubule force across chromosomes and thereby contributes to chromosome segregation synchrony (Matos et al., 2009). To analyze the role of small-number effects, future experiments could make use of single-molecule imaging or test the effect of cohesin complexes that can resist higher force because DNA gates are covalently closed (Haering et al., 2008; Richeldi et al., 2024). It is often stated that synchronous chromosome separation is important for proper genome inheritance, but, overall, our results suggest that a small amount of separation asynchrony is unavoidable and is well tolerated.

## Materials and methods

### *S. pombe* strains

All strains are listed in Table S2. *S. pombe* strains with the following modifications and mutations have been described previously: dh1L(cen1)-tdTomato, cnt1-GFP, cnt2-tdTomato (Sakuno et al., 2009), cen2-GFP (Yamamoto and Hiraoka, 2003), *cut9-665* and *cut1-206* (Hirano et al., 1986), *cut2-GFP*, securin overexpression (*natNT2:P.adh1(#6)-cut2-GFP:kanR*), and the inducible version of nondegradable cyclin B (*leu1+::P.nmt81-cdc13Δ(1-67)*) (Kamenz and Hauf, 2014). To fluorescently label the region close to the centromere of chromosome III, the plasmid

pSR14 was used to integrate a ~224x*tetO* array 556 bp 5′ of *meu27* following the previously described method (Rohner et al., 2008). In a similar manner, to fluorescently label the central centromere of chromosome III, the plasmid pSR13 was used to integrate a ~248x*lacO* array 2,369 bp 3′ of the start of the central centromere (Rohner et al., 2008). For inducible expression of nondegradable securin (ΔN-securin), the coding sequence of Cut2 lacking the first 75 amino acids was cloned into the pDUAL vector (Matsuyama et al., 2004) under the control of the *nmt81* promoter (Basi et al., 1993) and integrated at the *leu1* locus. This is the same construct as Cut2Δ80 (Funabiki et al., 1996). The discrepancy in numbers arises because an in-frame ATG sequence five amino acids upstream of the canonical start codon was previously interpreted as the start codon. For deletion of endogenous *klp5*, its coding sequence was replaced with an *hphMX* resistance cassette using PCR-based deletion methods for *S. pombe* (Bähler et al., 1998). For separase overexpression, *cut1* was cloned into a pDUAL vector (Matsuyama et al., 2004) under the control of the *cut1* promoter and integrated at the *leu1* locus, or the endogenous *cut1* promoter was replaced with an *ark1* promoter.

## Cell cultures and live-cell imaging

Prior to imaging, cells were cultured either in rich medium (yeast extract + adenine) or Edinburgh minimal medium (EMM) with the necessary supplements (Moreno et al., 1991). Cells were cultured at 30°C except for the strains carrying the temperature-sensitive alleles *cut9-665* or *cut1-206*, which were cultured at 25°C and incubated at 30°C for 30 min prior to imaging at 30°C. We used cells grown in EMM for the proteasome inhibition because this led to a more effective inhibition, while the *cut9-665* and *cut1-206* phenotypes were more pronounced after growth in rich medium. The *nmt81* promoter was repressed by the addition of 16 µM thiamine to EMM and induced by transferring the cells into EMM without thiamine for 14–18 h. Imaging was generally performed in EMM, except for the *cut9-665* strains and *cut1-206* strains, which were imaged in YEA (alternatively or additionally). Cells were mounted in lectin-coated (35–50 µg/ml, L1395; Sigma-Aldrich) culture dishes (8-well, Ibidi) and preincubated on the microscope stage at 30°C for 30 min. For partial inhibition of the proteasome, Bortezomib (B-1408; Velcade, LC Laboratories) was added to a final concentration of 100 µM. For partial destabilization of microtubules, 3, 5, or 10 µg/ml of carbendazim (MBC) was added 20–30 min before imaging. Live-cell imaging was carried out at 30°C on a DeltaVision system (Applied Precision/GE Healthcare) equipped with a climate chamber (EMBL) and a pco.edge 4.2 sCMOS camera, using a 60×/1.4 Apo oil objective (Olympus). GFP imaging used 461–489 nm LED illumination and a 525/48 emission filter; tdTomato imaging used 529–556 nm LED illumination and a 597/45 emission filter. Images were acquired using the "optical axis integration" modus of the softWoRx software over a range of 4 µm, which creates the equivalent of a sum projection of the imaged volume. To measure the time difference between the separation of two chromosomes, images were acquired every 3, 3.5 or 7 s for 1 h. To visualize securin-GFP dynamics, images were acquired every 15 s for 1.5–2 h.

## Data processing and analysis

Images were deconvolved using softWoRx software when necessary to improve signal clarity. The time point of sister chromatid separation was scored manually and was defined as the last time point at which sister chromatids moved coordinately or convergently before moving consistently toward opposing spindle poles. Kymographs were assembled using a custom MATLAB (RRID:SCR_001622) script, and the contrast was enhanced for easier visualization of the separation events. Trackmate (Ershov et al., 2022; Tinevez et al., 2017) with manual corrections was used in ImageJ (RRID:SCR_003070) for tracking the kinetics of sister chromatid separation, and custom MATLAB (RRID:SCR_001622) and R (RRID:SCR_001905) scripts were used to further process and display the data.

Time-series trajectories for tracked sister chromatid separation distances were generated in R. Distances were summarized at each time point by the cell population mean and standard deviation. To reduce frame-to-frame noise, mean trajectories were smoothed using generalized additive models with cubic regression splines. For plots where all three chromosomes are shown together, chromosome II traces were merged from both strains, using inverse-variance weighting when per-time point standard deviations were defined and, otherwise, by averaging the per-strain means.

To determine the kinetics of securin-GFP degradation, fluorescent intensities were quantified from time-lapse images. The nuclear signal of TetR-tdTomato was used to define the nucleus as a region of interest (ROI), and the average GFP signal intensity within the ROI was determined for each time point. The extracellular background was determined by averaging the signal intensity of three ROIs placed outside of the cell and was subtracted from the GFP signal. When more than one ROI was present (e.g., after nuclear division), their average was calculated before subtraction. For quantitative feature extraction from the degradation curves, local slopes were calculated by taking a derivative over seven consecutive points of a smoothed spline (the time point ± 3 frames). The onset of securin degradation was defined as the time point before this local slope repeatedly (>5 frames) dropped below 20% of a reference slope, which was calculated at 50% GFP intensity. The curves were normalized by setting the GFP level at degradation onset to 100% and the minimum of the smoothed curve to 0%. The normalized degradation rate was then approximated from the linear decay between 60 and 40% signal. The percentage of securin left at sister chromatid separation, the time between degradation onset and sister chromatid separation, and time points at which 90% of securin had been degraded were calculated using the normalized curves.

## Statistical analysis

Statistical analysis of the experimental data was performed using R, and results are listed in Table S1. The Gaussian distribution for each data set was generated using R's built-in normal density function (dnorm). Histograms were generated using a fixed bin width of 7 s, and y-axes were normalized to percent of cells. Datasets were compared by a two-sample Kolmogorov–Smirnov test unless stated otherwise.

## Deterministic model for separase release with positive feedback on securin degradation

In this model, securin (Sec) and separase (Sep) reversibly form a complex (SecSep), the APC/C activity increases from 0 to a positive value, and Sec is regulated by APC/C-mediated degradation. The APC/C activity is assumed to be zero in the basal state ($k_{APC/C} = 0$), and the SecSep complex is assumed to be in equilibrium with the free proteins. Sec is assumed to be in excess over Sep prior to anaphase (Kamenz et al., 2015), and the amount of free Sep is assumed to be negligible, i.e., the total concentrations of Sec ($Sec_{tot}$) and Sep ($Sep_{tot}$) are assumed to be much larger than the dissociation constant of the complex ($K_D = k_{off}/k_{on}$). Hence, the initial equilibrium can be approximated as

$$
\begin{aligned}
Sec &\approx Sec_{tot} - Sep_{tot} \\
Sep &\approx 0 \\
SecSep &\approx Sep_{tot}.
\end{aligned}
\quad (1)
$$

Sep, once released from Sec, accelerates the degradation of Sec. The differential equations describing this scenario are given by

$$
\begin{aligned}
\frac{dSec}{dt} &= -k_{on} \cdot Sec \cdot Sep + k_{off} \cdot SecSep - k_{APC/C} \cdot Sec \cdot \left( k_{basal} \right.\\
&\left. + k_{FB} \frac{Sep^h}{Sep^h + K_{FB,50}{}^h} \right)\\
\frac{dSep}{dt} &= -k_{on} \cdot Sec \cdot Sep + k_{off} \cdot SecSep + k_{APC/C} \cdot SecSep \cdot \left( k_{basal} \right.\\
&\left. + k_{FB} \frac{Sep^h}{Sep^h + K_{FB,50}{}^h} \right)\\
\frac{dSecSep}{dt} &= k_{on} \cdot Sec \cdot Sep - k_{off} \cdot SecSep - k_{APC/C} \cdot SecSep\\
&\cdot \left( k_{basal} + k_{FB} \frac{Sep^h}{Sep^h + K_{FB,50}{}^h} \right).
\end{aligned}
\quad (2)
$$

The positive feedback (FB) is assumed to be nonlinear and is described using the Hill equation.

Numerical simulations were performed using the ODE system in Eq. 2. The initial steady state was calculated by setting $k_{APC/C} = 0$. Anaphase was simulated by assuming a sigmoidal increase in $k_{APC/C}$ activity

$$
k_{APC/C}(t) \approx k_{APC/C,max} \cdot \frac{t^n}{t^n + t_{50}{}^n}.
\quad (3)
$$

The following parameter values were assumed in Fig. 3 D: total Sep concentration $Sep_{tot} = 0.05\ \mu M$; total Sec concentration $Sec_{tot} = 0.1\ \mu M$; $k_{APC/C,max} = 0.02\ s^{-1}$; $k_{on} = 1\ \mu M^{-1}\ s^{-1}$; $k_{off} = 10^{-4}\ s^{-1}$; $t_{50} = 100\ s$; n = 10; $k_{basal} = 1$; $k_{FB} = 10$ (dotted lines) or $k_{FB} = 0$ (solid lines); $K_{FB,50} = 0.02\ \mu M$; h = 2.

## Deterministic model for separase release with separase autoactivation

In this model, free Sep, once released from Sec, performs auto-cleavage to generate a highly active Sep species ($Sep_{active}$), which is more efficient in auto- and cohesin cleavage. The differential equations describing this scenario are given by

$$
\begin{aligned}
\frac{dSec}{dt} &= -k_{on} \cdot Sec \cdot Sep + k_{off} \cdot SecSep - k_{APC/C} \cdot Sec \\
\frac{dSep}{dt} &= -k_{on} \cdot Sec \cdot Sep + k_{off} \cdot SecSep + k_{APC/C} \\
&\quad \cdot SecSep - \left( k_{a,1} \cdot Sep^h + k_{a,2} \cdot Sep_{active}{}^h \right) \cdot Sep \\
\frac{dSecSep}{dt} &= k_{on} \cdot Sec \cdot Sep - k_{off} \cdot SecSep - k_{APC/C} \cdot SecSep \\
\frac{dSep_{active}}{dt} &= \left( k_{a,1} \cdot Sep^h + k_{a,2} \cdot Sep_{active}{}^h \right) \cdot Sep.
\end{aligned}
\quad (4)
$$

Auto-cleavage occurs with an exponent $h$ that prevents premature amplification and may arise from oligomerization of separase.

The APC/C activity ($k_{APC/C}$) and the irreversible cleavage reactions ($k_{a,1}$, $k_{a,2}$) are assumed to be zero in the basal state. The basal concentrations of Sec, Sep, and the SecSep complex are the same as in Eq. 1.

Anaphase was simulated by assuming a sigmoidal increase in APC/C activity (Eq. 3). The following parameter values were assumed in Fig. 4 B and Fig. S4 G: total Sep concentration $Sep_{tot} = 0.05\ \mu M$; total Sec concentration $Sec_{tot} = 0.1\ \mu M$; $k_{APC/C,max} = 0.02\ s^{-1}$ (WT) or $0.0066\ s^{-1}$ (APC/C mut); $k_{on} = 10^{-3}\ \mu M^{-1}\ s^{-1}$; $k_{off} = 10^{-7}\ s^{-1}$; $t_{50} = 100\ s$; $n = 10$; $k_{a,1} = 0.1$; $k_{a,2} = 300$; $h = 2$.

## Stochastic model for cohesin cleavage

The model assumes that the $i$-th chromosome begins with an initial number of cohesin complexes, $N_i$, and its sister chromatids separate once its cohesin count decays to a threshold $n_i$. Cohesin cleavage is modeled as an inhomogeneous Poisson process. The rate constant of cohesin cleavage is assumed to be identical across all chromosomes and to increase gradually over a timeframe $\tau$, reflecting the gradual release and/or activation of separase in the cell (Eq. 5).

$$
k(t) = \begin{cases} k_{max} \dfrac{t}{\tau}, & 0 \le t \le \tau, \\[2mm] k_{max}, & t > \tau. \end{cases}
\quad (5)
$$

The primary output of the model is the distribution of two pairwise separation-time differences, $T_{12} = T_1 - T_2$ and $T_{32} = T_3 - T_2$, where $T_i$ denotes the separation time for chromosome $i$.

The perturbed cases are assumed to undergo the following parameter changes:

### MBC treatment
Destabilization of microtubules by MBC is expected to reduce the pulling force on sister chromatids and thereby cause a decrease in the threshold number of cohesins that are required to hold sister chromatids together. Since this effect applies to all three chromosomes, a common multiplicative factor, $\alpha < 1$, is applied to all three threshold values, $\{n_1, n_2, n_3\}$.

### Separase mutation
Mutated separase is expected to have lower catalytic activity, corresponding to a lower $k_{max}$. To represent this effect, a multiplicative factor, $\beta_k < 1$, is applied to $k_{max}$.

### APC/C mutation
Mutated APC/C is expected to degrade securin more slowly and elongate the ramp phase for separase activity. To represent this effect, a multiplicative factor, $\beta_\tau > 1$, is applied to $\tau$.

### Velcade treatment

Velcade has a similar effect to the APC/C mutation. To represent this effect, a multiplicative factor, $\beta_{\tau 2} > 1$, is applied to $\tau$.

Additionally, we implemented two mechanistic variants that encode different hypotheses about cohesin cleavage:

### Processive separase action

This model variant assumes processive cohesin cleavage with a fixed burst size $b$ (number of cohesins cleaved and removed within one event).

### Steric hindrance

This model variant considers steric hindrance by packed chromosomes that lowers accessibility of separase molecules to cohesins in the interior of chromosomes. For simplicity, we assume that cohesin complexes are distributed in a sphere and that only those at the surface are accessible. As outer cohesin complexes get cleaved, inner complexes are exposed. Under these assumptions, the effective cleavage rate scales with the surface-to-volume ratio of the region occupied by the remaining cohesin. Assuming a uniform density of cohesin in the sphere, the volume of the sphere scales with the number of remaining cohesin complexes, i.e., $V \sim N$. Consequently, the surface-to-volume ratio scales as $\frac{A}{V} \sim V^{-\frac{1}{3}} \sim N^{-\frac{1}{3}}$. Therefore, in this model variant, we assume (Eq. 6)

$$k_{eff}(N, t) = \begin{cases} k(t), & N \leq n_{inner}, \\ k(t)\left(\dfrac{N}{n_{inner}}\right)^{-1/3}, & N > n_{inner}, \end{cases} \tag{6}$$

where $n_{inner}$ is the number of cohesins accommodated by the innermost core, a region with no more steric hindrance effect.

Additionally, to evaluate if positive feedback causes a sharp increase in separase activity (Fig. 4 B), we further considered a variation of the three models above, in which the time duration for the rate increase, $\tau$, is constrained at low values ($\tau < 5\ s$).

The six models resulting from the combinations above are summarized in Table S3. They were each fitted to the experimental data (WT and perturbations). The bounds for parameter fitting are listed in Table S4.

All simulations and optimization procedures were implemented in Python using standard scientific libraries. Computationally intensive parameter fitting and cross-validation were performed in parallel on a high-performance computing cluster. Parameter estimates are reported from best-fit solutions with uncertainties derived from cross-validated ensembles, and model performance was summarized by the mean cross-validated aggregate EMD.

### Gillespie stochastic simulation

To simulate cohesin cleavage, we implemented a modified Gillespie algorithm. Each simulation tracks the number of cohesin complexes on each of the chromosomes and records the time when each chromosome reaches its corresponding cohesin count threshold for separation.

To accommodate the time-dependent degradation rate constant (Eq. 5), we modified the formula for sampling the next-reaction time from the classical Gillespie formula, $\Delta T = T_{next} - T_{prev} = -\frac{\ln(r)}{k_{max}N_{total}}$, to (Eq. 7)

$$\Delta T = T_{next} - T_{prev} = \sqrt{-\frac{2\tau \cdot \ln(r)}{k_{max}N_{total}} + T_{prev}^2} - T_{prev}, \tag{7}$$

where $T_{next}$ is the next degradation time to be sampled, $T_{prev}$ is the previous degradation time, $N_{total}$ is the sum of the number of cohesin complexes currently remaining on the three chromosomes, and $r$ is a uniform random number between 0 and 1. After sampling for the next cleavage time, the cohesin cleavage is randomly chosen to happen to the $i$-th chromosome with probability $N_i/N_{total}$. For the processive separase action model, every cleavage event removes $b$ cohesin complexes on the chosen chromosome. For the steric hindrance model, $N_{total}$ is modified to the sum of the currently accessible number of cohesin complexes on all chromosomes, where the accessible number is $N_{i,access} = N_i \cdot \left(N_i/n_{inner}\right)^{-\frac{1}{3}}$ for $N_i > n_{inner}$ and $N_{i,access} = N_i$ for $N_i \leq n_{inner}$; chromosome selection probability is also modified to $N_{i,access}/N_{total,access}$.

Note that Eq. 7 gives the inverse-CDF sampling for the next-reaction time with CDF, $P(\Delta T < t) = 1 - \exp(-k_{max}N_{total}[(T_{prev} + t)^2 - T_{prev}^2]/2\tau)$, which can be derived from Eq. 5 through $P(\Delta T < t) = 1 - \exp(-\int_{T_{prev}}^{T_{prev}+t} N_{total}k(t')dt')$. Beyond the ramp phase, the regular Gillespie algorithm (Gillespie, 1976; Gillespie, 1977) based on rate constant $k_{max}$ was used.

The Gillespie simulation was used to generate sample time trajectories (Fig. 5 F) and to benchmark the fast simulation method.

### Fast simulation

To enable efficient parameter optimization and large-scale validation, we simulated the cohesin degradation dynamics in three chromosomes separately (based on their mutual independence) and adopted alternative exact simulation methods, using either order statistics (models without steric hindrance) or vectorized sum of waiting times approach (model with steric hindrance). These methods are statistically exact, alike the Gillespie algorithm, but offer considerable computational speedups (100×–1000×), allowing scalable model fitting.

### Accelerated sampling via order statistics

For the basic model, the cleavages of individual cohesins are independent events following an identical waiting time distribution with the following CDF:

$$P(T_{wait} < t) = \begin{cases} 1 - \exp\left(-\dfrac{k_{max}t^2}{2\tau}\right), & 0 \leq t \leq \tau, \\ 1 - \exp\left(-\dfrac{k_{max}\tau}{2} - k_{max}(t - \tau)\right), & t > \tau. \end{cases} \tag{8}$$

The separation time for the $i$-th chromosome is the $(n_i+1)$-th largest cohesin cleavage time out of a total of $N_i$ cleavage times that follow the distribution in Eq. 8. As each cleavage time can be converted from a uniform random number through CDF inverse, which is a monotonically increasing function, the $(n_i+1)$-th largest cleavage time is the CDF inverse of the $(n_i+1)$-th largest

value of $N_i$ uniform random numbers. The latter is known to follow the beta distribution $Beta(N_i-n_i,n_i+1)$. Therefore, by directly drawing one random number from this beta distribution and converting it through the inverse of Eq. 8, we obtain a sample separation time of a chromosome with a highly efficient $O(1)$ operation.

The accelerated sampling method applies to the processive separase action model, too, when cohesin complexes are treated as predefined groups. A single cleavage event eliminates one group of $b$ cohesins. To utilize the order statistics framework, we transform the parameters as effective counts,

$$\widehat{N}_i = \text{ceil}\left(\frac{N_i}{b}\right), \widehat{n}_i = \widehat{N}_i - \text{ceil}\left(\frac{N_i - n_i}{b}\right). \quad (9)$$

The process then proceeds as the non-processive case with $\widehat{N}_i$ and $\widehat{n}_i$.

### Vectorized simulation
For the steric hindrance model, the effective rate constant depends on the system's current state (Eq. 6). Therefore, cleavage events of individual cohesin complexes are no longer independent of each other, making the fast-sampling method invalid. To accelerate the simulation, we take advantage of the convenient feature that there is only one reaction in the system and employ vectorization across simulations. Instead of simulating one trajectory at a time, we simultaneously simulate $M$ independent cohesin loss trajectories. For each step from state $N$ to $N–1$, we use Eq. 7 to determine the next event times across all $M$ simulations. This approach leverages efficient array operations to achieve a significant speedup.

### Parameter estimation and model selection
For each candidate parameter set, the simulation methods described above generated a batch of $M = 10{,}000$ independent samples of $T_{12}$ and $T_{32}$, respectively, for each experimental scenario. Model fit was assessed using the EMD (Earth Mover's Distance, Wasserstein-1 distance) (Bazán et al., 2019; Rubner et al., 1998), a robust goodness of fit metric that quantifies the similarity between the empirical distribution of experimental data and that of the simulated data.

The objective function was defined as the summed EMD values across five experiments (control, MBC, separase mutant, APC/C mutant, and velcade), which is also termed aggregate EMD. Sample size of 10,000 was confirmed to generate a sufficiently precise estimate of EMD (Fig. S5 D).

Model parameters were estimated using a two-stage optimization protocol. First, a global search was performed using differential evolution, an efficient population-based global optimization algorithm that explores the parameter space for the global optimum (Storn and Price, 1997). The algorithm was implemented using the `scipy.optimize.differential_evolution()` function in the SciPy library. We used the "best1bin" strategy with a population size of 10 (corresponding to 10× the number of parameters in each model variant), mutation factor (0.5, 1.0), recombination constant 0.7, and a relative convergence tolerance of 0.01. The top-performing candidate solutions were then refined locally using the L-BFGS-B algorithm (Byrd et al., 1995) to maximize fit quality. Choices of population size and convergence tolerance were informed by benchmark results shown in Fig. S5, E and F.

In addition to the global optimization to identify the optimal parameters, fivefold cross-validation was performed to reduce bias and avoid overfitting. Specifically, experimental data were split into five subsets (folds). In five iterations, model parameter optimization was conducted on four folds (80% of the data), and aggregate EMD was computed on the held-out fold (20% of data). Through this process, five fitted parameter sets (Fig. 5 C and D) were obtained for each model variant and compared for consistency. The fivefold cross-validation results are also reported in Fig. S5 E and F, whereas all other stochastic model-related figures are based on the optimal parameter set obtained through fitting the model to all data.

### Parameter sensitivity analysis
To evaluate the impact of each parameter on separation synchrony, we performed a One-at-a-time (OAT) sensitivity analysis. Each parameter was perturbed across the range of this parameter while holding all others fixed at the previously determined optimal value. For each parameter set, we ran 10 batches of 10,000 separation time difference ($\Delta t$) simulations.

### Online supplemental material
Fig. S1 shows additional experiments that characterize securin degradation and sister chromatid segregation in WT cells; Fig. S2 provides more details for the separase mutant data shown in Fig. 2; Fig. S3 similarly provides more details for the securin and separase overexpression data in Fig. 2; Fig. S4 provides (1) more detailed data for the MBC treatment shown in Fig. 2, (2) demonstrates that *klp5* deletion did not greatly influence chromosome separation synchrony, (3) provides additional simulation data for the separase autoactivation model that distinguishes between free separase and active separase, (4) shows the securin degradation kinetics for the *cut9-665* APC/C mutant and velcade-treated cells, and (5) shows a kymograph of sister chromatid separation in a velcade-treated cell; Fig. S5 shows additional simulation results from the stochastic model. Table S1 lists the statistical analyses conducted; Table S2 lists the *S. pombe* strains used; Table S3 provides an overview of the stochastic model variants and their parameters; Table S4 lists the parameters used in the stochastic models, as well as the parameter boundaries used for fitting and the rationale for those.

## AI-assisted code generation for stochastic models
Stochastic modeling, optimization, and analysis scripts were developed using Google Gemini 3.1 Pro (Antigravity IDE). Initial stochastic simulation and optimization code was written by K.P.; AI was used to expand upon and modify the code (including additional model variants, optimization, and analysis scripts). Final code and outputs were reviewed and validated by K.P., W.W., and J.C.

## Data availability

The data underlying Δt distributions in Figs. 1, 2, 3, 4, and 5 are openly available in Zenodo (RRID:SCR_004129) at https://doi.org/10.5281/zenodo.19462276. The code for the stochastic model is openly available in Zenodo at https://doi.org/10.5281/zenodo.19434950. Yeast strains are available from S.H. (silke@vt.edu).

## Acknowledgments

We thank Saahil Golia and Tatiana Boluarte for help with strain construction and image analysis; Douglas Weidemann for scripts; Takeshi Sakuno and Yoshinori Watanabe (University of Tokyo, Japan), and Yasushi Hiraoka (Osaka University, Japan) for strains; the National BioResource Project (NBRP), Japan, for sending us strain FY15657; Sarah Gilmour and Sarah Zanders (Stowers Institute for Medical Research, Kansas City, MO, USA) for long-read sequencing and ddPCR of centromere-tagged strains; Eric de Sturler for help with early stages of the stochastic model; as well as the Boehringer Ingelheim Fonds (fellowship to J. Kamenz).

Research reported in this publication was supported by the National Institute of General Medical Sciences of the National Institutes of Health under award numbers R35GM119723 (S. Hauf), R35GM149565 (S. Hauf), and R35GM138370 (J. Chen). Open Access funding provided by the Max Planck Society.

Author contributions: Wendi Williams: formal analysis, investigation, software, validation, visualization, and writing—original draft. Kien Phan: formal analysis, methodology, software, validation, and writing—review and editing. Jing Chen: formal analysis, funding acquisition, methodology, resources, software, supervision, validation, and writing—review and editing. Stefan Legewie: formal analysis and writing—review and editing. Julia Kamenz: conceptualization, formal analysis, investigation, methodology, visualization, and writing—original draft, review, and editing. Silke Hauf: conceptualization, formal analysis, funding acquisition, investigation, supervision, and writing—original draft.

Disclosures: The authors declare no competing interests exist.

Submitted: 13 February 2026

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

# Supplemental material

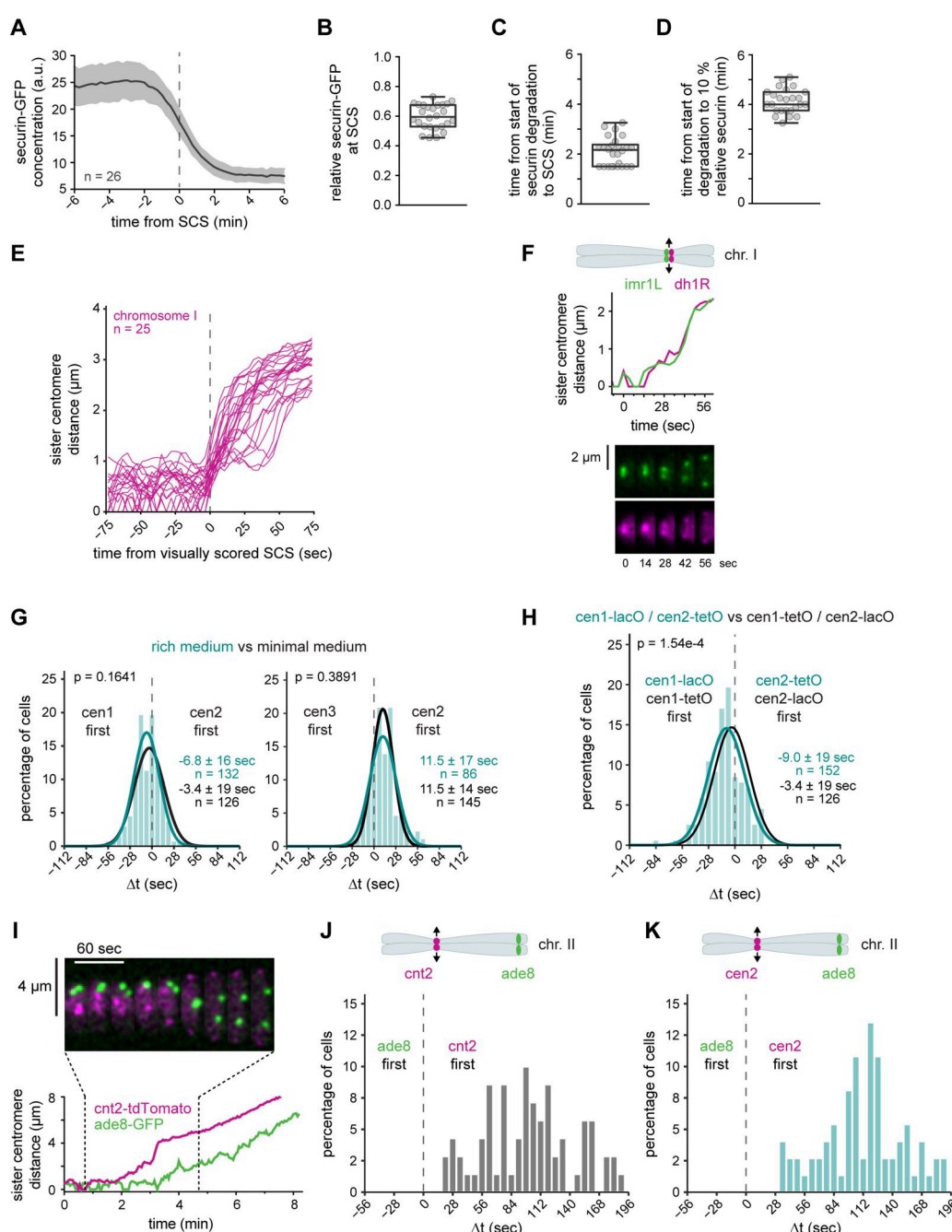

Figure S1.  **Dynamics of securin degradation and separation of centromeres and chromosome arms. (A)** The nuclear securin (Cut2)-GFP abundance of WT cells was followed by live-cell imaging. The cen1-tdTomato marker was used to determine sister chromatid separation (SCS), and individual time courses were aligned to this point (vertical dashed line). Mean (line) ± SD (shaded area) of the cell population; n = number of cells. **(B)** Quantification of the amount of securin-GFP present at the time of sister chromatid separation (SCS) relative to the amount present at the start of degradation. Data are from the experiment shown in A. Single-cell measurements are as circles; the boxplot shows median, interquartile range (box), and range (whiskers). **(C and D)** Quantification of the time from the start of securin-GFP degradation until SCS (C) or until 90% of securin-GFP had been degraded (D). Same representation as described in B. **(E)** Distances between sister centromeres of chromosome I in a subset of WT cells (same experiment as in Fig. 1 F). Curves are aligned to sister chromatid separation (SCS) determined by visual inspection of the time-lapse recording. **(F)** Sister centromere distance and corresponding kymograph for a strain with dh1R-tdTomato and imr1L-GFP markers on the same chromosome. **(G)** Cyan: Frequency distributions and Gaussian fit (continuous lines) of the time difference (Δt) between the separation of outer centromere markers on chromosomes I and II or chromosomes II and III for cells grown in rich medium. The fitted Gaussian distributions of cells grown in minimal medium (black) are shown for comparison. Mean ± SD of the fit; n = number of cells; P values from a two-sample Kolmogorov–Smirnov test. **(H)** Cyan: Frequency distribution and Gaussian fit (continuous line) of the time difference (Δt) between the separation of outer centromere markers on chromosomes I and II or chromosomes II and III using cen1-*lacO* and cen2-*tetO* (flipped *lacO/tetO* relative to the standard tagging). The fitted Gaussian distribution of cells with the standard tagging scheme (cen1-*tetO*, cen2-*lacO*, the same as in Fig. 1 F) is shown in black for comparison. Mean ± SD of the fit; n = number of cells; P values from a two-sample Kolmogorov–Smirnov test. **(I)** Sister centromere distance and corresponding kymograph for a cell from J with chromosome II arm marker (ade8-GFP) and cnt2-tdTomato. **(J and K)** Frequency distributions of the time difference between the separation of a chromosome II arm marker (ade8-GFP) and either cnt2-tdTomato (J) or cen2-tdTomato (K).

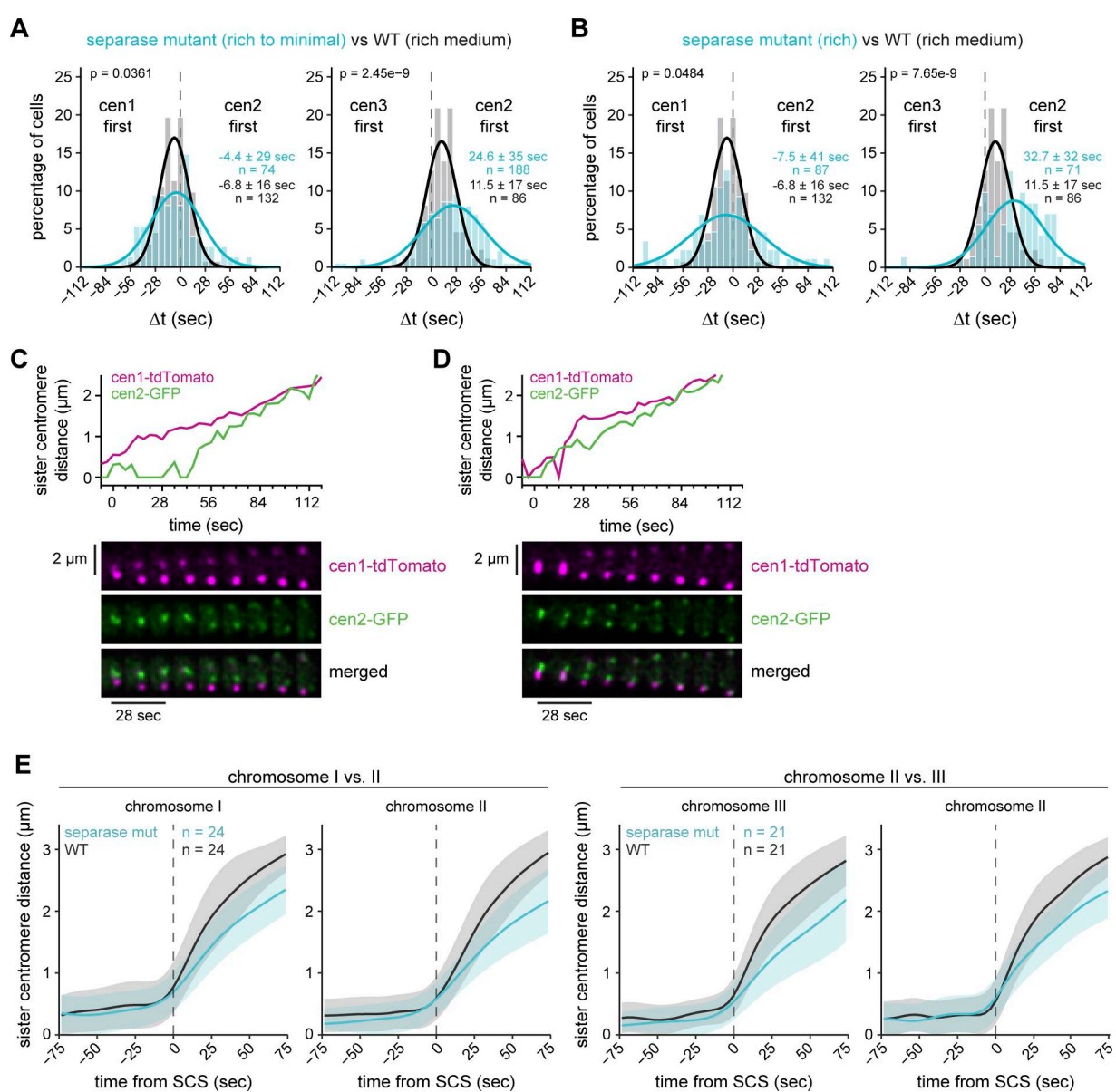

Figure S2. **Lower separase activity reduces centromere separation synchrony and speed. (A and B)** Cyan: Frequency distributions and Gaussian fit (continuous lines) of the time difference between the separation of centromeres 1 and 2 or centromeres 2 and 3 for cells carrying the temperature-sensitive separase allele *cut1-206*, grown in rich medium and then imaged in minimal medium (A) or rich medium (B). The fitted Gaussian distributions of WT cells (black) are shown for comparison. Mean ± SD of the fit; n = number of cells; P values from a two-sample Kolmogorov–Smirnov test. Combined separase mutant datasets from A and B are shown in Fig. 2 A, as distributions were not significantly different (Table S1). **(C and D)** Sister centromere distance and corresponding kymographs for *cut1-206* cells with cen1-tdTomato and cen2-GFP markers. Examples for asynchronous (C) and more synchronous separation (D). **(E)** Distance between each sister centromere pair in WT and separase-mutant cells (same cells as shown in Fig. 2 B). Distances are aligned to sister chromatid separation (SCS) of the respective chromosome at t = 0. Mean (line) ± SD (shaded area) of the cell population; n = number of cells.

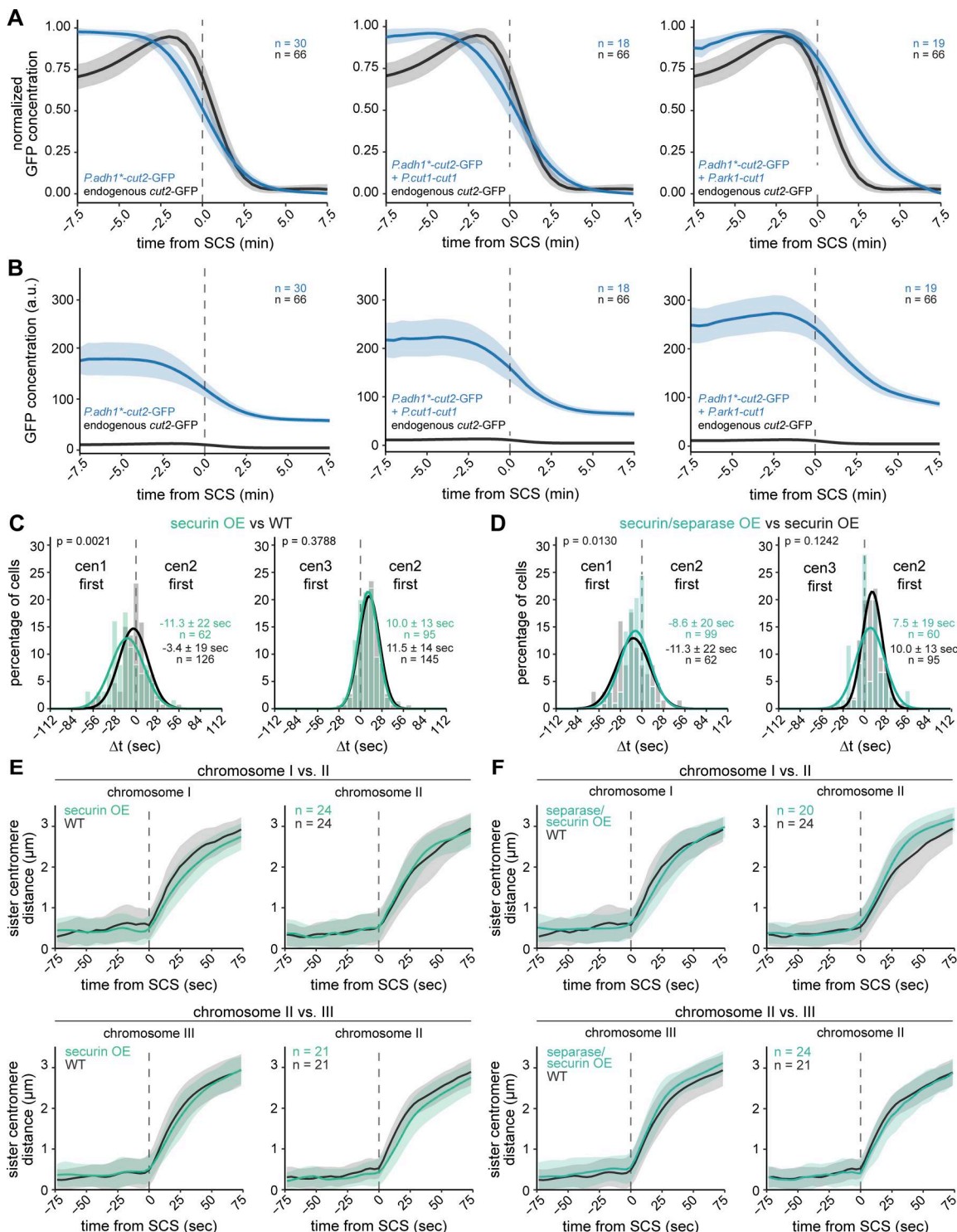

Figure S3. **Co-overexpression of securin and separase does not strongly change chromosome separation synchrony. (A and B)** Normalized (A) and non-normalized (B) concentration measurements of securin (Cut2)-GFP in strains expressing endogenous *cut2*-GFP (black) or overexpressing *cut2*-GFP from the endogenous locus using a mutated *adh1* promoter (*P.adh1\**, blue). Cut2-GFP is either overexpressed alone (left panel) or co-overexpressed with separase, Cut1. For separase overexpression, either a second copy with the endogenous *cut1* promoter was integrated at the *leu1* locus (*P.cut1*, center panel) or the *cut1* promoter at the endogenous locus was replaced with the *ark1* promoter (*P.ark1*, right panel). Mean (line) ± SD (shaded area) of the cell population; n = number of cells. **(C and D)** Frequency distributions and Gaussian fit (continuous lines) of the time difference between the separation of centromeres 1 and 2 or centromeres 2 and 3 for cells with securin overexpression alone (C), or securin co-overexpression with separase (*P.cut1-cut1*) (D); same experiment as in Fig. 2 C. The fitted Gaussian distributions of WT cells (C) or cells with securin overexpression alone (D) are shown for comparison in black. Mean ± SD of the fit; n = number of cells; P values from a two-sample Kolmogorov–Smirnov test. **(E and F)** Distance between each sister centromere pair in WT cells and cells with securin overexpression (E) or securin and separase co-overexpression (F); same cells as shown in Fig. 2 D. Distances are aligned to sister chromatid separation (SCS) of the respective chromosome at t = 0. Mean (line) ± SD (shaded area) of the cell population; n = number of cells.

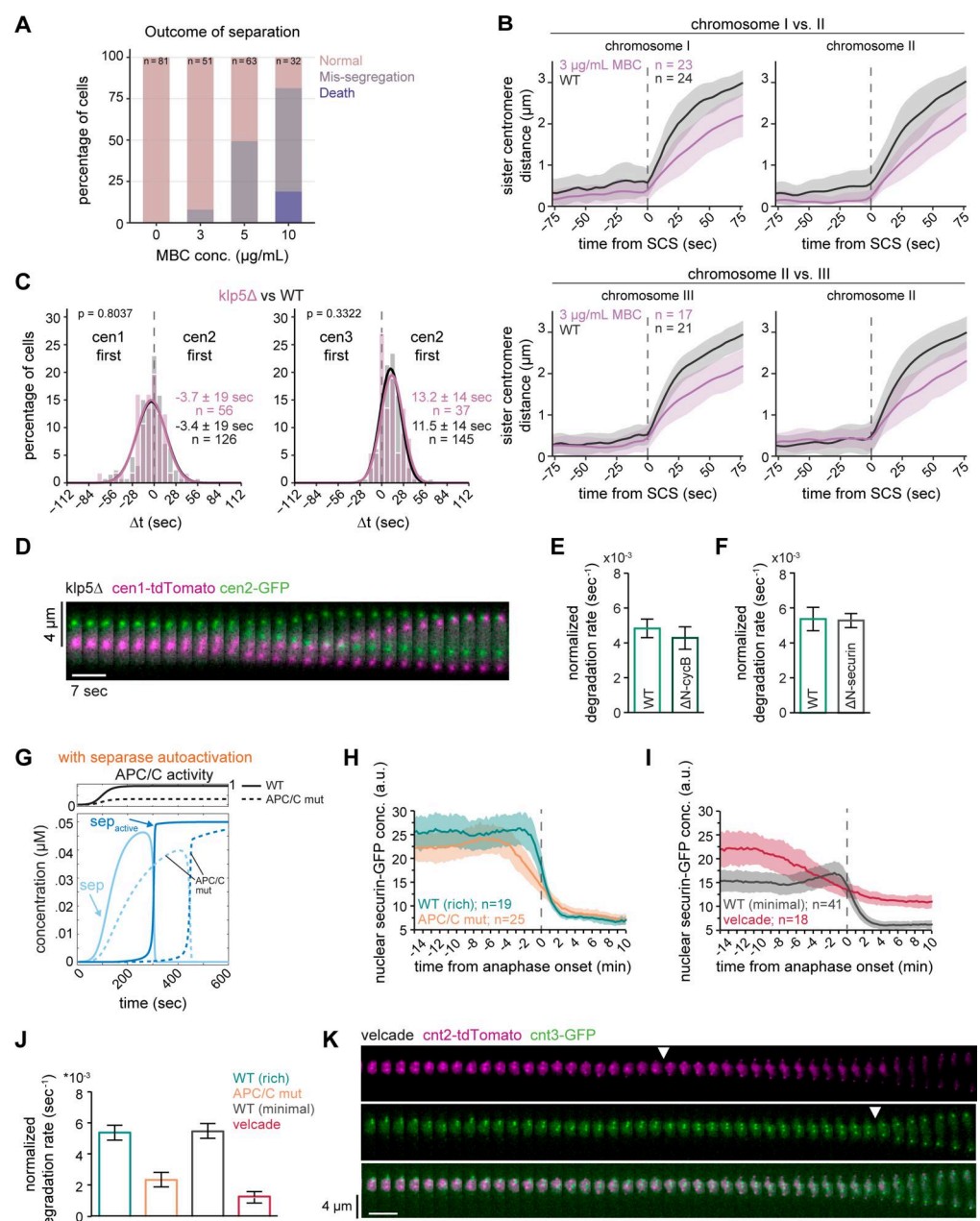

Figure S4. **Additional analyses of the influence of microtubule perturbations and securin degradation dynamics on chromosome segregation synchrony. (A)** Anaphase outcomes of cells that underwent chromosome segregation in medium containing different concentrations of MBC, shown as the percentage of all cells segregating. Outcomes were defined as either normal segregation, mis-segregation, or cell death. **(B)** Distance between each sister centromere pair in WT cells and cells with 3 µg/ml MBC; same cells as shown in Fig. 2 F. Distances are aligned to sister chromatid separation (SCS) of the respective chromosome at t = 0. Mean (line) ± SD (shaded area) of the cell population; n = number of cells. **(C)** Frequency distributions and Gaussian fit (continuous lines) of the time difference between the separation of centromeres 1 and 2 or centromeres 2 and 3 for cells with the klp5 gene deleted (klp5Δ). The fitted Gaussian distributions of WT cells (black) are shown for comparison. Mean ± SD of the fit; n = number of cells; P values from a two-sample Kolmogorov–Smirnov test. **(D)** Representative kymograph of sister chromatid separation in a klp5Δ cell with cen1-tdTomato and cen2-GFP markers, illustrating the mis-aligned chromosomes in metaphase. **(E)** Quantification of the securin degradation rates in the presence (ΔN-cycB, n = 22) or absence (WT, n = 31) of nondegradable cyclin B (Cdc13) (mean and SD). The corresponding securin degradation curves have been shown previously by Kamenz and Hauf (2014), Fig. 1 E. **(F)** Quantification of the securin degradation rates in the presence (ΔN-securin, n = 21) or absence (WT, n = 27) of nondegradable securin (mean and SD). The corresponding securin degradation curves are shown in Fig. 3 E. **(G)** Simulation of released separase (light blue) and active separase (darker blue) from the model with separase auto-activation (shown in Fig. 4 B) assuming high (solid lines) or low (dashed lines) APC/C activity. See Materials and methods for details. **(H)** Nuclear securin(Cut2)-GFP intensity was monitored by live-cell imaging of WT cells and cut9-665 temperature-sensitive mutants (APC/C mut). Anaphase onset was determined using the cen1-tdTomato marker, and individual time courses were aligned to this point (vertical dashed line). Mean (line) ± SD (shaded area) of the cell population; n = number of cells. **(I)** Nuclear securin(Cut2)-GFP intensity was monitored by live-cell imaging in WT cells grown in minimal medium, or cells additionally treated with 100 µM of the proteasome inhibitor velcade (bortezomib). Mean (line) ± SD (shaded area) of the cell population; n = number of cells. **(J)** Quantification of the degradation rates for the experiments shown in H and I after normalization of the data to the minimum and maximum values. Error bars represent the SD. **(K)** Kymograph for a velcade-treated cell with cnt2-tdTomato and cnt3-GFP markers. Arrowheads indicate sister centromere separation.

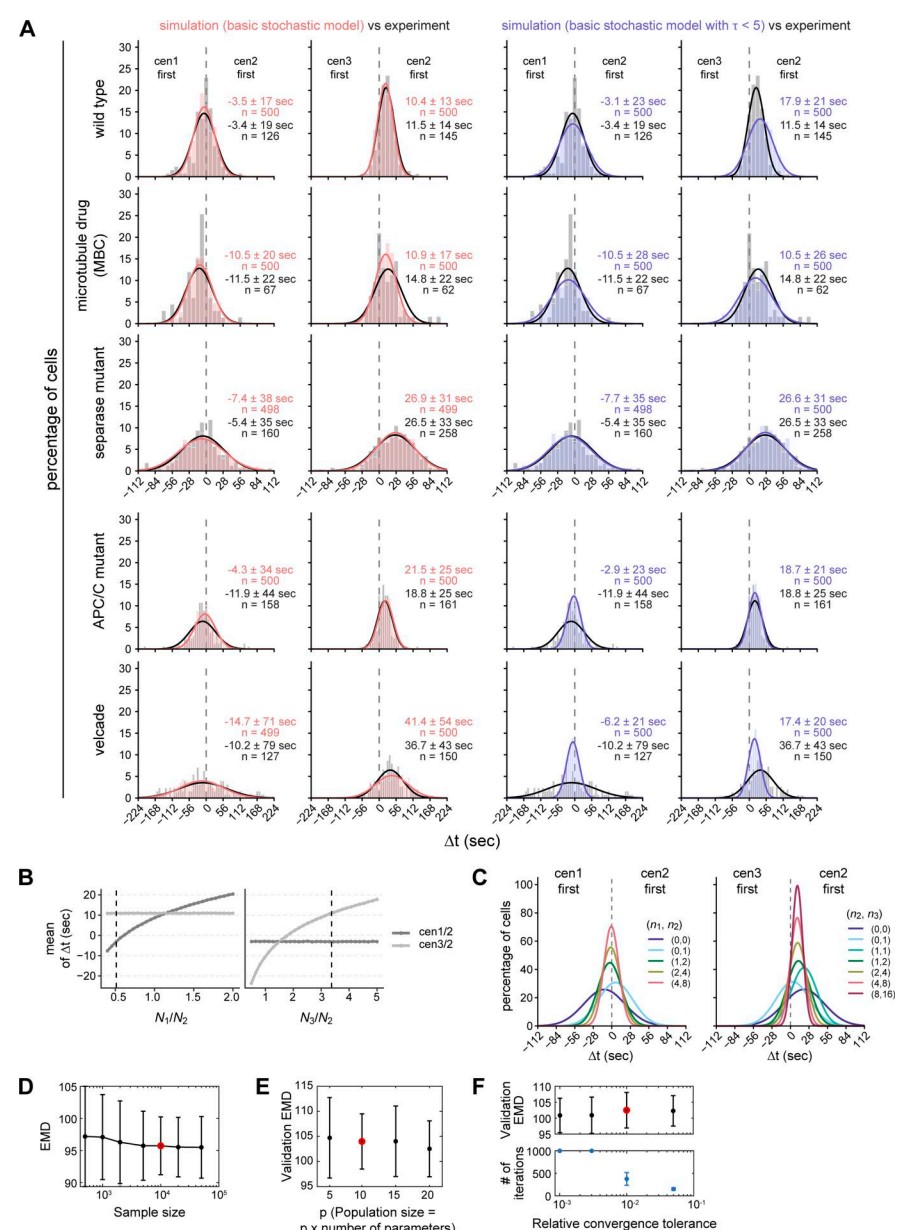

Figure S5. **Simulation of sister chromatid separation timing using stochastic models. (A)** Experimental data compared with stochastic model simulations of differences in sister chromatid separation times using the basic stochastic model from Fig. 5 or the basic stochastic model with a restriction on the value of τ (τ < 5 s). Frequency distributions and Gaussian fit (continuous lines) of the time difference between the separation of centromeres 1 and 2 or centromeres 2 and 3 are shown for stochastic simulations of WT cells, cells treated with 3 µg/ml MBC, cells carrying the temperature-sensitive separase allele *cut1-206* (separase mutant), cells carrying a temperature-sensitive allele of the APC/C subunit Cut9 (*cut9-665*, APC/C mutant), and cells exposed to 100 µM velcade (bortezomib). The frequency distributions (gray) and fitted Gaussian distributions (black) of the experimental data are shown for comparison. Mean ± SD of the simulation data or the Gaussian fit (for experimental data); n = number of cells in the experimental data or simulation. **(B)** One-at-a-time (OAT) sensitivity analysis of the basic stochastic model showing the effect of relative cohesin load on mean separation timing. The ratios of initial cohesin numbers between chromosome pairs ($N_1/N_2$ and $N_3/N_2$) were varied individually while all other parameters were held fixed at their optimal values. The resulting mean of the time difference (Δt) between the separation of centromeres 1 and 2 or 2 and 3 is shown. Dashed vertical lines indicate the optimal parameter values. **(C)** Effect of low cohesin thresholds on separation time variability in the basic model. Gaussian fits of the time difference between the separation of centromeres 1 and 2 or 2 and 3 are shown for simulations in which the separation threshold n was set to specific values for each chromosome, denoted by ($n_1$, $n_2$) or ($n_2$, $n_3$), while all other parameters were held constant at their optimal values. **(D)** EMD estimate is unbiased, exhibiting a stable mean across sample sizes, and its precision (standard error) stabilizes beyond a sample size of 10,000. The optimal parameter set fitted to the complete experimental dataset was used, yielding lower EMD values than the fivefold cross-validation EMDs shown in E and F. Mean ± SD of five repeats. **(E)** Varying the population size in differential evolution has no significant effect on the optimization results. We therefore used P = 10 in this study. Mean ± SD of fivefold cross-validation EMD values are shown. **(F)** Mean and SD of fivefold cross-validation EMD are stable across the tested range of relative tolerances (upper panel). However, because stochastic simulations are limited by finite sampling, convergence is constrained by simulation sample size. Consequently, stringent relative tolerances (<0.01) trigger spurious non-convergence warnings and incur unnecessary computational cost (lower panel; maximum iteration limit reached for the two smallest tolerances) after the practical convergence limit has already been achieved. We therefore selected a relative tolerance of 0.01. Error bars represent mean ± SE. Red circles indicate the hyperparameter values chosen in this study.

Provided online are Table S1, Table S2, Table S3, and Table S4. Table S1 shows statistical analysis of experimental results. Table S2 shows *S. pombe* strains. Table S3 shows stochastic model variants. Table S4 shows model parameters and their bounds for fitting.

