## [Peer Review File · The Journal of Cell Biology]

Chromosome segregation synchrony in *S. pombe* is noise-limited and arises without positive feedback

Wendi Williams, Kien Phan, Jing Chen, Stefan Legewie, Julia Kamenz, and Silke Hauf

Corresponding Author(s): Silke Hauf, Virginia Tech and Julia Kamenz, University of Groningen

Review Timeline:

Submission Date:	2026-02-13
Editorial Decision:	2026-03-24
Revision Received:	2026-04-07

Monitoring Editor: Arshad Desai

Scientific Editor: Gabriele Stephan

Transaction Report:

DOI: <https://doi.org/10.1083/jcb.202602088>

March 24, 2026

RE: JCB Manuscript #202602088

Silke Hauf
Virginia Tech

Dear Prof. Hauf:

Thank you for submitting your manuscript entitled "Chromosome segregation synchrony in *S. pombe* is noise-limited and arises without positive feedback". Your paper has been assessed by three expert reviewers. All of the reviewers found your analysis and conclusions persuasive and of significance in understanding the anaphase transition. While Reviewer #1 recommends publication of the submitted manuscript, Reviewers #2 and #3 provided feedback that should be addressable via text changes to the manuscript. We would thus be interested in receiving a revision that addresses the reviewer comments.

A. MANUSCRIPT ORGANIZATION AND FORMATTING:

Full guidelines are available on our Instructions for Authors page, <http://jcb.rupress.org/submission-guidelines#revised>.

- 1) Text limits: Character count for Articles is < 40,000, not including spaces. Count includes abstract, introduction, results, discussion, and acknowledgments. Count does not include title page, figure legends, materials and methods, references, tables, or supplemental legends.
- 2) Figures limits: Articles may have up to 10 main text figures.
- 3) Figure formatting: Scale bars must be present on all microscopy images, including inset magnifications. Molecular weight or nucleic acid size markers must be included on all gel electrophoresis. Aspect ratios of images may not be altered.
- 4) Statistical analysis: Error bars on graphic representations of numerical data must be clearly described in the figure legend. The number of independent data points (n) represented in a graph must be indicated in the legend. Statistical methods should be explained in full in the materials and methods. For figures presenting pooled data the statistical measure should be defined in the figure legends. Please also be sure to indicate the statistical tests used in each of your experiments (either in the figure legend itself or in a separate methods section) as well as the parameters of the test (for example, if you ran a t-test, please indicate if it was one- or two-sided, etc.). Also, if you used parametric tests, please indicate if the data distribution was tested for normality (and if so, how). If not, you must state something to the effect that "Data distribution was assumed to be normal but this was not formally tested."
- 5) Abstract and title: The abstract should be no longer than 160 words and should communicate the significance of the paper for a general audience. The title should be less than 100 characters including spaces. Make the title concise but accessible to a general readership.
- 6) Materials and methods: Should be comprehensive and not simply reference a previous publication for details on how an experiment was performed. Please provide full descriptions in the text for readers who may not have access to referenced manuscripts.
- 7) All antibodies, cell lines, animals, and tools used in the manuscript should be described in full, including accession numbers for materials available in a public repository such as the Resource Identification Portal. Please be sure to provide the sequences for all of your primers/oligos and RNAi constructs in the materials and methods. You must also indicate in the methods the source, species, and catalog numbers (where appropriate) for all of your antibodies. Please also indicate the acquisition and quantification methods for immunoblotting/western blots.
- 8) Microscope image acquisition: The following information must be provided about the acquisition and processing of images:
 - a. Make and model of microscope
 - b. Type, magnification, and numerical aperture of the objective lenses
 - c. Temperature
 - d. Imaging medium
 - e. Fluorochromes
 - f. Camera make and model
 - g. Acquisition software

h. Any software used for image processing subsequent to data acquisition. Please include details and types of operations involved (e.g., type of deconvolution, 3D reconstitutions, surface or volume rendering, gamma adjustments, etc.).

10) Supplemental materials: There are strict limits on the allowable amount of supplemental data. Articles may have up to 5 supplemental figures. Please also note that tables, like figures, should be provided as individual, editable files. A summary of all supplemental material should appear at the end of the Materials and methods section.

**11) eTOC summary: A ~40-50-word summary that describes the context and significance of the findings for a general readership should be included on the title page. The statement should be written in the present tense and refer to the work in the third person.

**13) ORCID IDs: ORCID IDs are unique identifiers allowing researchers to create a record of their various scholarly contributions in a single place. Please note that ORCID IDs are now *required* for all authors. At resubmission of your final files, please be sure to provide your ORCID ID and those of all co-authors.

Please note that JCB now requires authors to submit Source Data used to generate figures containing gels and Western blots with all revised manuscripts. This Source Data consists of fully uncropped and unprocessed images for each gel/blot displayed in the main and supplemental figures. For assays performed using capillary electrophoresis and/or immunoassay-based detection, authors should instead provide the electropherogram graph(s) for each experiment, plotting fluorescence/chemiluminescence intensity vs. molecular weight/size. Please be sure to provide one Source Data file for each figure gels, blots, and/or capillary electrophoresis assays along with your revised manuscript files. File names for Source Data figures should be alphanumeric without any spaces or special characters (i.e., SourceDataF#, where F# refers to the associated main figure number or SourceDataFS# for those associated with Supplementary figures). For traditional gels and blots, the lanes of the gels/blots should be labeled as they are in the associated figure, the place where cropping was applied should be marked (with a box), and molecular weight/size standards should be labeled wherever possible. For capillary electrophoresis assays, each trace in the graph should be color-coded and labeled to indicate which protein, gene, or sample is being measured (please try to avoid red/green combinations to accommodate our color-blind readers).

Journal of Cell Biology now requires a data availability statement for all research article submissions. These statements will be published in the article directly above the Acknowledgments. The statement should address all data underlying the research presented in the manuscript. Please visit the JCB instructions for authors for guidelines and examples of statements at (<https://rupress.org/jcb/pages/editorial-policies#data-availability-statement>).

B. FINAL FILES:

-- Cover images: If you have any striking images related to this story, we would be happy to consider them for inclusion on the journal cover. Submitted images may also be chosen for highlighting on the journal table of contents or JCB homepage carousel.

Images should be uploaded as TIFF or EPS files and must be at least 300 dpi resolution.

****It is JCB policy that if requested, original data images must be made available to the editors. Failure to provide original images upon request will result in unavoidable delays in publication. Please ensure that you have access to all original data images prior to final submission.****

****The license to publish form must be signed before your manuscript can be sent to production. A link to the license to publish form will be sent to the corresponding author only. Please take a moment to check your funder requirements before choosing the appropriate license.****

Thank you for your attention to these final processing requirements. Please revise and format the manuscript and upload materials within 7 days. If you need an extension for whatever reason, please let us know and we can work with you to determine a suitable revision period.

Thank you for this interesting contribution, we look forward to publishing your paper in Journal of Cell Biology.

Sincerely,

Arshad Desai
Monitoring Editor
Journal of Cell Biology

Gabriele Stephan
Scientific Editor
Journal of Cell Biology

Reviewer #1:

This is a beautiful, comprehensive study that illustrates how computer modelling can synergise with experimental manipulation to test and exclude hypotheses to define a mechanism. I congratulate the authors and have no revisions to suggest.

Reviewer #2:

In this manuscript, entitled "Chromosome segregation synchrony in *S. pombe* is noise-limited and arises without positive feedback," the authors combine high-resolution live-cell imaging with computational modelling to investigate the mechanisms underlying the synchrony of sister chromatid separation in fission yeast.

At the onset of anaphase, sister chromatids separate abruptly and irreversibly following the cleavage of cohesin complexes, which topologically entrap the two chromatids, by the protease separase. Across a range of organisms, from yeast to humans, sister chromatid separation has been observed to occur synchronously and sharply. However, it is not known whether such a switch-like increase in separase activity results from positive feedback regulation, which is a common feature of major cell cycle transitions.

Using fission yeast as a model system, the authors report that anaphase synchrony primarily depends on the rapid degradation of the separase inhibitor securin, leading to separase activation, and does not require separase-mediated positive feedback. This conclusion is supported by both experimental observations and computational modelling. Furthermore, the authors propose that the observed separation dynamics can be purely explained by a stochastic model of cohesion cleavage.

In addition to the convincing data, the paper is very well written. Therefore, I have only a few (minor) comments:

Minor comments:

1. Page 6: "However, this change was considerably less than that observed in the separase mutant, which-while having similarly slow centromere movement during anaphase-had standard deviations of 35 sec for cen1 vs. cen2 and 34 sec for cen2 vs. cen3

- ($p = 4.78e-9$ and $1.17e-7$, respectively, by Levene's test) (Fig. 2B,F)". This is not shown in Fig. 2B,F but rather in Fig. 2A.
2. Page 6-7, for non-degradable cyclin B (Delta-N-cyclin B) and non-degradable securin (Delta-N-securin) constructs used in this study, could the authors clarify the exact sequence range deleted from the N terminus of cyclin B or securin?
 3. Fig.4B, I suggest adding an explanation for dashed and solid lines to the figure legend. Although it is clear that this a model in which APC/C activity is reduced it could help to add some information to the legend for clarity reasons.
 4. In the reference, 'Rohner, S., S.M. Gasser, and P. Meister. 2008. Modules for cloning-free chromatin tagging in *Saccharomyces cerevisiae*. *Yeast*. 25:235-239. doi:10.1002/yea.1580.' There is a typo for "cerevisae".
 5. In the reference: Unsworth, A. et al. 2008, doi is missing.

Reviewer #3:

In this manuscript by Williams et al., the authors set out to study the mechanisms that drive synchronous chromosome segregation in anaphase. Prior findings in budding yeast identified a positive feedback mechanism for anaphase onset, although it is unclear whether such mechanism exists in other systems or not. Using a combination of live imaging microscopy, mechanistic analyses, and mathematical modeling, the authors demonstrate that fission yeast chromosomes segregate in a mostly synchronized manner, although there is a bias in the order, with chromosome 1 segregating first, followed by 2 and 3. To test whether this synchrony is dependent on positive feedback, the authors tested different models and found that none of them could explain the observed synchrony of chromosome segregation. Instead, the authors favor a model whereby the irreversible nature of cohesin cleavage by separase, limited by molecular noise, is sufficient to explain synchronous nature of chromosome segregation without the need of a positive feedback mechanism.

The data are very convincing, and I am particularly impressed by the elegant combination of detailed microscopy analyses with mathematical modeling. I only have two concerns regarding the data in Figure 5. First, the authors estimate the number of cohesin molecules at the start of chromosome segregation for each chromosome that explain the bias in segregation order, but after reading the text and the methods I am not sure I fully understand where these numbers come from. It would greatly help the manuscript if the authors could provide more details as to how fitting the basic model to the experimental data led to those numbers. Second, the authors refer to "molecular noise" in the title and the abstract, but in the text they refer to "small-number effects". Are these two terminologies the same? If so, it would be helpful if the authors stuck to one term throughout, to avoid confusing non-experts. If the authors could clarify these two points, I would be happy to support the publication of this manuscript at *The Journal of Cell Biology*.

We are grateful to the reviewers for their assessment of our manuscript. We have implemented the suggested text changes.

In addition, we have

- made minor changes to figure legends for consistency.
- have fused Supplemental Figures S4, S5, and S6 into a new Figure S4 to comply with JCB's requirement of a maximum of five supplemental figures.
- made minor changes to Figure S3 (same y-axis for all panels in Fig. S3B).
- made the labeling in what is now Figure S4G (model 2) consistent with the main figure (Fig. 4B).
- added information on the microscopy setup and AI usage.
- rounded the p-values in Table S1 for consistency with figures.

Reviewer #1:

This is a beautiful, comprehensive study that illustrates how computer modelling can synergise with experimental manipulation to test and exclude hypotheses to define a mechanism. I congratulate the authors and have no revisions to suggest.

We appreciate the positive assessment. Thank you.

Reviewer #2:

In this manuscript, entitled "Chromosome segregation synchrony in *S. pombe* is noise-limited and arises without positive feedback," the authors combine high-resolution live-cell imaging with computational modelling to investigate the mechanisms underlying the synchrony of sister chromatid separation in fission yeast.

At the onset of anaphase, sister chromatids separate abruptly and irreversibly following the cleavage of cohesin complexes, which topologically entrap the two chromatids, by the protease separase. Across a range of organisms, from yeast to humans, sister chromatid separation has been observed to occur synchronously and sharply. However, it is not known whether such a switch-like increase in separase activity results from positive feedback regulation, which is a common feature of major cell cycle transitions.

Using fission yeast as a model system, the authors report that anaphase synchrony primarily depends on the rapid degradation of the separase inhibitor securin, leading to separase activation, and does not require separase-mediated positive feedback. This conclusion is supported by both experimental observations and computational modelling. Furthermore, the authors propose that the observed separation dynamics can be purely explained by a stochastic model of cohesion cleavage.

In addition to the convincing data, the paper is very well written. Therefore, I have only a few (minor) comments:

Minor comments:

1. Page 6: "However, this change was considerably less than that observed in the separase mutant, which-while having similarly slow centromere movement during anaphase-had standard deviations of 35

sec for cen1 vs. cen2 and 34 sec for cen2 vs. cen3 ($p = 4.78e-9$ and $1.17e-7$, respectively, by Levene's test) (Fig. 2B,F)". This is not shown in Fig. 2B,F but rather in Fig. 2A.

We thank the reviewer for pointing this out. We have moved the Fig. 2B reference to the part of the sentence that concerns speed of movement and have added the correct Fig. 2A reference at the end.

The corrected sentence reads: However, this change was considerably less than that observed in the separate mutant, which—while having similarly slow centromere movement during anaphase (Fig. 2B)—had standard deviations of 35 sec for cen1 vs. cen2 and 34 sec for cen2 vs. cen3 ($p = 4.78e-9$ and $1.17e-7$, respectively, by Levene's test) (Fig. 2A).

2. Page 6-7, for non-degradable cyclin B (Delta-N-cyclin B) and non-degradable securin (Delta-N-securin) constructs used in this study, could the authors clarify the exact sequence range deleted from the N terminus of cyclin B or securin?

Thank you for this suggestion. The sequence deletions are listed in Supplementary Table S2 with the strain genotypes. However, to make the information more easily accessible, we have added this information to the main text and specifically indicate that Δ N-cyclin B lacks residues 1-67, and Δ N-securin lacks residues 1-75. The same Δ N-securin construct was previously called Δ 1-80 (Funabiki *et al.*, 1996), based on an outdated start codon annotation, which we now mention in the Methods section.

3. Fig.4B, I suggest adding an explanation for dashed and solid lines to the figure legend. Although it is clear that this a model in which APC/C activity is reduced it could help to add some information to the legend for clarity reasons.

Thank you for the suggestion. We have added to the Fig. 4B legend: "Solid lines indicate high APC/C activity; dashed lines indicate low APC/C activity, mimicking an APC/C mutant."

4. In the reference, 'Rohner, S., S.M. Gasser, and P. Meister. 2008. Modules for cloning-free chromatin tagging in *Saccharomyces cerevisiae*. *Yeast*. 25:235-239. doi:10.1002/yea.1580.' There is a typo for "cerevisae".

Funnily, this typo is present in the paper and in PubMed. Since fixing it might interfere with automated searches, we have kept the typo.

5. In the reference: Unsworth, A. et al. 2008, doi is missing.

Thank you for noticing. The doi has been added.

Reviewer #3:

In this manuscript by Williams et al., the authors set out to study the mechanisms that drive synchronous chromosome segregation in anaphase. Prior findings in budding yeast identified a positive feedback mechanism for anaphase onset, although it is unclear whether such mechanism exists in other systems or not. Using a combination of live imaging microscopy, mechanistic analyses, and mathematical modeling, the authors demonstrate that fission yeast chromosomes segregate in a mostly synchronized manner, although there is a bias in the order, with chromosome 1 segregating first, followed by 2 and 3. To test whether this synchrony is dependent on positive feedback, the authors tested different models and found that none of them could explain the observed synchrony of chromosome segregation. Instead, the authors favor a model whereby the irreversible nature of cohesin cleavage by separase, limited by

molecular noise, is sufficient to explain synchronous nature of chromosome segregation without the need of a positive feedback mechanism.

The data are very convincing, and I am particularly impressed by the elegant combination of detailed microscopy analyses with mathematical modeling. I only have two concerns regarding the data in Figure 5. First, the authors estimate the number of cohesin molecules at the start of chromosome segregation for each chromosome that explain the bias in segregation order, but after reading the text and the methods I am not sure I fully understand where these numbers come from. It would greatly help the manuscript if the authors could provide more details as to how fitting the basic model to the experimental data led to those numbers.

Thank you for pointing out that this is not sufficiently clear.

In brief, the number of cohesin molecules initially present on each chromosome (N_1 , N_2 , N_3) are parameters being estimated. We first set boundaries for the minimum and maximum possible numbers, based on information in the literature (Supplementary Table S4), and then estimated the values within this range by fitting the model to the data.

We set a minimum of 50 and a maximum of 1,000 cohesin molecules for chromosome 2—or, more precisely, for the centromere of chromosome 2, which we consider the relevant region, given the position of our markers. The numbers for chromosome 1 and chromosome 3 are implemented in the model relative to chromosome 2 through the model parameters R_{12} and R_{32} , which are the ratios N_1/N_2 and N_3/N_2 . The number for chromosome 1 can range between 0.4 and 2-times that of chromosome 2; the one for chromosome 3 can range between 0.5- and 5-times that of chromosome 3 (Table S4); i.e. we allow the numbers to be either smaller or larger, and we were rather generous in allowing for a larger number on chromosome 3, given its later separation.

After fitting the model to the data, we obtained the numbers shown in Fig. 5D. Those were well within the ranges provided ($N_2 \sim 430$, $R_{12} \sim 0.5$, $R_{32} \sim 4.0$), indicating that the model was not constrained by the limits we had set.

The detailed fitting procedure (described in the Supplemental Text) may be too much information for the main text, but we have expanded the main text to hopefully explain this better. We now specifically state that N_1 , N_2 , and N_3 are parameters being estimated through fitting the model to the data.

We hope we have interpreted the reviewer's request correctly. If not, please let us know.

Second, the authors refer to "molecular noise" in the title and the abstract, but in the text they refer to "small-number effects". Are these two terminologies the same? If so, it would be helpful if the authors stuck to one term throughout, to avoid confusing non-experts. If the authors could clarify these two points, I would be happy to support the publication of this manuscript at The Journal of Cell Biology.

"Molecular noise" and "small-number effects" refer to the same phenomenon, but to slightly different aspects; "noise" is the broader term, "small-number effects" specifies the precise mechanism. We introduced the "noise" terminology to meet the character limitations in the title.

In more detail, our model predicts that the very small number of cohesin molecules remaining at the time of separation leads to stochasticity in separation order, limiting synchrony. So, our broader conclusion is that molecular noise (stochastic fluctuations due to the inherently stochastic nature of biochemical processes) naturally limits synchrony.

We agree that this distinction was not clear. We have now attempted to mitigate this by adding a sentence at the end of the introduction to clarify the relationship between the two terms.